



# A 500-year runoff reconstruction for European catchments

Sadaf Nasreen[1], Markéta Součková[1], Mijael Rodrigo Vargas Godoy[1], Ujjwal Singh[1], Yannis Markonis[1], Rohini Kumar[2], Oldrich Rakovec[1,2], and Martin Hanel[1]

[1]Faculty of Environmental Sciences, Czech University of Life Sciences Prague, Praha-Suchdol 16500, Czech Republic
[2]UFZ-Helmholtz Centre for Environmental Research, Leipzig 04318, Germany

**Correspondence:** Martin Hanel (hanel@fzp.czu.cz)

**Abstract.** Since the beginning of this century, Europe has been experiencing severe drought events (2003, 2007, 2010, 2018 and 2019) which have had an adverse impacts on various sectors, such as agriculture, forestry, water management, health, and ecosystems. During the last few decades, projections of the impact of climate change on hydroclimatic extremes were often capable of reproducing changes in the characteristics of these extremes. Recently, the research interest has been extended
to include reconstructions of hydro-climatic conditions, so as to provide historical context for present and future extremes. While there are available reconstructions of temperature, precipitation, drought indicators, or the $20^{th}$ century runoff for Europe, long-term runoff reconstructions are still lacking (e.g, monthly or daily runoff series for short periods are commonly available). Therefore, we considered reconstructed precipitation and temperature fields for the period between 1500 and 2000 together with reconstructed scPDSI, natural proxy data, and observed runoff over 14 European catchments to calibrate and
validate the semi-empirical hydrological model GR1A and two data-driven models (Bayesian recurrent and long short-term memory neural network). The validation of input precipitation fields revealed an underestimation of the variance across most of Europe. On the other hand, the data-driven models have been proven to correct this bias in many cases, unlike the semi-empirical hydrological model GR1A. The comparison to observed historical runoff data has shown a good match between the reconstructed and observed runoff and between the runoff characteristics, particularly deficit volumes. The reconstructed runoff
is available via figshare, an open source scientific data repository under the DOI https://doi.org/10.6084/m9.figshare.15178107, (Sadaf et al., 2021).





## 1 Introduction

Global warming has impacted numerous land surface processes (Reinecke et al., 2021) over the last few decades, resulting in
more severe droughts, heat waves, floods, and other extreme weather. Droughts, in particular, pose a serious threat to Europe's
hydrology. The flow of many rivers is greatly hampered by prolonged droughts, which restrain the availability of fresh water
for agriculture and domestic use. For example, the 2003 drought significantly reduced European river flows by approximately
60 to 80% relative to the average. Likewise, the annual flow levels of several river gauges have decreased by 9 to 22% over
the last decade (Krysanova et al., 2008; Middelkoop et al., 2001; Uehlinger et al., 2009; Su et al., 2020) due to a lack of
rainfall and a warmer climate. For the last 40 years, low river flows have rendered water-power generation impossible in
the UK, resulting in a 45£ million loss each year. However, there has been less focus on the water deficit in streams, rivers
and other reservoir's, the so-called hydrological droughts (Van Loon, 2015). Most importantly, runoff, which supplies rivers
with a significant amount of water, is potentially valuable for water security management. The challenging element is that
the nonlinear behavior of hydro-climate fluctuations cannot be explicitly interpreted using data from the most recent centuries
(Markonis and Koutsoyiannis, 2016). Continuous records of runoff/discharge series are no longer available, including various
multi-year droughts and pluvial periods. On the other hand, proxy-based (typically seasonal or annual) reconstructions are
alternatively used, considering various proxy data, such as past tree-rings (Cook et al., 2015; Casas-Gómez et al., 2020; Tejedor
et al., 2016; Kress et al., 2010; Nicault et al., 2008), speleothem (Vansteenberge et al., 2016), ice cores, sediments (Luoto and
Nevalainen, 2017) and documentary and instrumental evidence (Pfister et al., 1999; Dobrovolný et al., 2010; Wetter et al.,
2011; Brázdil and Dobrovolný, 2009) to pinpoint extreme events and the detection of climate change.

The majority of existing reconstructions focus on temperature (Dobrovolný et al., 2010; Luterbacher et al., 2004; Emile-
Geay et al., 2017; Trouet et al., 2013; Büntgen et al., 2006; Luterbacher et al., 2004; Casty et al., 2005; Moberg et al., 2008;
Xoplaki et al., 2005), precipitation (Boch and Spötl, 2011; Wilson et al., 2005; Murphy et al., 2018; Wilhelm et al., 2012) or
drought (Büntgen et al., 2010; Brázdil et al., 2018; Cook et al., 2015; Kress et al., 2014; Tejedor et al., 2016; Ionita et al.,
2017; Hanel et al., 2018) and flood reconstructions (Swierczynski et al., 2012; Wetter et al., 2011). A few studies have been
conducted for the reconstruction of runoff-drought deficit series (Hanel et al., 2018; Moravec et al., 2019; Hansson et al., 2011;
Kress et al., 2014; Martínez-Sifuentes et al., 2020). However these studies are either local or regional, or cover a relatively
short time period. As an example of Hansson et al. (2011), which introduced a runoff series for the Baltic Sea only, between
1550 and 1995 using temperature and atmospheric circulation indices. Similarly, Sun et al. (2013) has used tree-ring proxies
to reconstruct runoff in the Fenhe River Basin in China's Shanxi region over the last 211 years.

Conversely, the reconstructed precipitation and temperature series (or fields) can be used to reconstruct runoff with a hydro-
logical model (Tshimanga et al., 2011; Armstrong et al., 2020). This can be achieved through a process-based model of varying
complexity, with the advantage of following general physical laws – e.g., preserving mass balance, etc. Physical based models:
MIKE SHE (Im et al., 2009) and VELMA (Laaha et al., 2017) or data-driven methods, such as support vector machines (Ji
et al., 2021; Zuo et al., 2020), artificial neural networks (ANNs; Kwak et al., 2020; Hu et al., 2018; Senthil Kumar et al.,
2005), random forests (Breiman, 2001; Contreras et al., 2021), and Shannon entropy (Thiesen et al., 2019) are able to capture


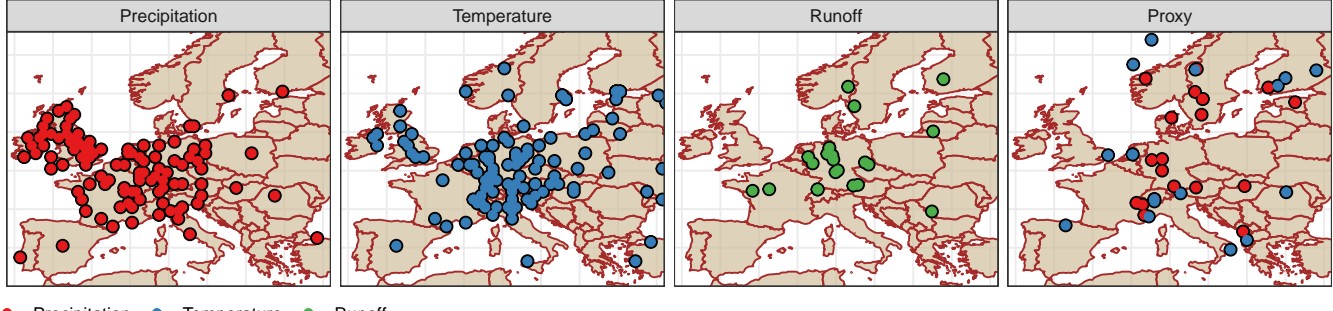

**Figure 1.** Spatial distribution of the observed GHCN precipitation and temperature stations, GRDC discharge gauges and proxies for precipitation and temperature.

complex non-linear relationships. While the lack of physical constraints in the data-driven models limits their application under contrasting (changing) boundary conditions (in comparison with those of the model training period), their advantage is that they can often directly use biased reconstructed data as an input series.

The objective of the present study is to provide a long-term, hydrological reconstruction for the Central European catchments, utilizing the available gridded precipitation (Pauling et al., 2006) and temperature (Luterbacher et al., 2004) reconstructions, natural proxies (Ljungqvist et al., 2016) and other long-term historical data sources. Specifically, we use a combination of a conceptual hydrological model (GR1A; Mouelhi et al., 2006) and two data-driven models (Chen et al., 2020; Okut, 2016) to simulate the annual evolution of runoff over the period 1500–2000. We pay particular attention to low flows during drought
years. Using long-term data on climatic conditions and runoff may provide an efficient technique of visualizing droughts and low flow periods. The structure of the paper is as follows: the considered hydroclimatic reconstructions, natural proxies and observed data are described in Section 2. In Section 3, we introduce the data selection and pre-processing, hydrological and data-driven models and the drought identification. The reconstructed input fields, as well as our runoff simulations considering four input data combinations (precipitation, temperature, raw proxy and drought indicator) and two data-driven approaches,
together with the hydrological model are evaluated in Section 4. Finally, we provide certain guidelines on the advantages and limitations of reconstructed datasets in the concluding section 5.

## 2   Data

Herein, we used precipitation (Pauling et al., 2006) and temperature (Luterbacher et al., 2004) reconstructions for the past half-millennium, data from the Old World Drought Atlas (Cook et al., 2015), and natural proxies (Ljungqvist et al., 2016).
For benchmarking, we relied on the observational data records of precipitation and temperature (Menne et al., 2018), as well as runoff from the Global Runoff Data Center (GRDC; Fekete et al. 1999). The data-sets are summarized in Table 1 and are described in more detail below.



**Table 1.** Summary of considered data sets

| Reference | Domain | Temporal coverage (CE) | Spatial resolution | Variables |
|---|---|---|---|---|
| Pauling et al. (2006) | Europe | 1500–2000 | 0.5° x 0.5° | Seasonal Precipitation |
| Luterbacher et al. (2004) | Europe | 1500–2000 | 0.5° x 0.5° | Seasonal Temperature |
| Menne et al. (2018) | Global | 1760–2010 | 26000 point stations | Mean Temperature |
| Menne et al. (2018) | Global | 1760–2010 | 20590 point stations | Mean Precipitation |
| Ljungqvist et al. (2016) | Northern Hemisphere | 800–2005 | 130 point stations | Temperature |
| Ljungqvist et al. (2016) | Northern Hemisphere | 800–2005 | 197 point stations | Hydro-proxies |
| Cook et al. (2015) | Europe | 0–2012 | 0.5° x 0.5° | Palmer Drought Severity Index |

## 2.1 Hydroclimatic reconstructions

We used reconstructed seasonal precipitation and temperature gridded data (0.5° x 0.5°) over Europe (30.25° N-70.75°N /
29.75°W-39.75°E) from 1500 to the present day. To this end Pauling et al. 2006, reconstructed precipitation ($P$) was done
by applying principal component regression to documented evidence (i.e., memoirs, annals, newspapers), speleothem proxy
records (Proctor et al., 2000) and tree-ring chronologies from the International Tree-Ring Data Bank (ITRDB). Reconstructed
temperature ($T$) is obtained from Luterbacher et al. (2004) which relies on historical records and seasonal natural proxies
(i.e., ice cores from Greenland and tree-rings from Scandinavia and Siberia). We refer to these data sets as reconstructed
forcings. Additionally, we used data from the Old World Drought Atlas (OWDA; Cook et al. 2015) which contains information
regarding moisture conditions across Europe, specifically the self-calibrated Palmer Drought Severity Index (scPDSI) using
summer-related, tree-ring proxies for the period from 0 to 2012 CE.

## 2.2 Other hydro-climate proxy information

We also included a raw proxy series for hydroclimatic variables by Ljungqvist et al. 2016 in our analysis, as mentioned in
Supplementary tables (S1 and S2). We considered 20 precipitation related proxies consisting of three tree-ring widths, eight
lake sediments, five peat bogs, two speleothems and two peat humidifications. Similarly, there were 17 temperature-based
proxies including six tree-rings, three ice cores, three lake sediments, two speleothems and three written records. These proxies
are not evenly distributed across Europe (Fig. 1). The available series, typically spanning hundreds of years, were restricted
to 1500 – 2000 in our study. Data standardization was conducting by subtracting the mean and dividing by standard deviation
(both calculated considering the time-series after 1900). Missing values were calculated by linear approximation and, in this
way, we obtained a consistent set of proxy information for each (annual) time step. It has been previously established that these
proxies correlate well with climatic variables, such as precipitation and temperature (Riechelmann and Gouw-Bouman, 2019).





## 2.3 The Global Historical Climatology Network (GHCN)

The GHCN dataset (GHCN; (Peterson and Vose, 1997)) – one of the largest observational databases, collated by the National
Oceanic and Atmospheric Administration (NOAA, Quayle et al., 1999) – was used to verify the accuracy of the precipitation
and temperature reconstructions. The GHCN-m (version 2) data-set contains observed temperature, rainfall and pressure data
from 1701 to 2010. Data for the majority of stations are, however, available after 1900. GHCN-m precipitation and temperature
from GHCN V2, as well as from the new GHCN V4 version were included in the preliminary analysis (Menne et al., 2012).
We found 113 precipitation and 144 temperature stations within the European domain (see Fig. 1) with records dating back
earlier than 1875. Most stations are geographically concentrated in Central Europe, and few stations are located in the eastern
and northern areas of Europe (see Fig. 2). These data, hereafter, are referred to as the GHCN data.

## 2.4 Observed runoff

The Global Runoff Data Center (GRDC; https://www.bafg.de/GRDC/EN/Home/homepage_node.html) provides data for more
than 2780 gauging stations in Europe, with the oldest records starting from 1806. The runoff series from the GRDC were
selected based on the condition of data availability, at least 25 years prior to 1900. In total, there were 21 such stations
predominantly available in Central Europe: 11 in Germany, two in France, two in Switzerland, one in the Czech Republic,
one in Sweden, one in Finland, one in Lithuania and one in Romania (see Fig. 1). These stations cover 12 European river
basins (Rhine, Loire, Elbe, Danube, Wesser, Main, Glama, Slazach, Nemunas, Gota Alv, Inn and Kokemaenjoke), with areas
ranging from nearly $6\,100$ km$^2$ (Kokemaenjoki, Muroleenkoski, Finland) to $576\,000$ km$^2$ (Danube, Orsova, Romania). The
mean annual discharge (Q$_{mean}$) varies from 50 m$^3$s$^{-1}$ to $5\,600$ m$^3$s$^{-1}$ and spans different time periods for each catchment.
The most extensive records were available in KRV Sweden and Dresden, containing the longest discharge series of 212 and
208 years, respectively. The gauging station in Köln also provided 195 years of data for the Rhine River. Note that some of the
gauging stations are located in close proximity and therefore have a greater degree of similarity in relation to the runoff time-
series (e.g., two stations in Basel, Rhine). Detailed information relating to all the selected stations and their silent characteristics
are provided in Table 2.

## 2.5 Study area

In the first section of the study, the analysis is performed across the European region bounded by (30.25° N-70.75°N / 29.75°W-
39.75°E), in which the grid-based reconstruction of precipitation and temperature was verified against the observation data.
In the second section, we focus on 21 specific Central European catchments, corresponding to the available long-term GRDC
discharge records. The study area and the observational data of the hydroclimatic variables are shown in Figure 2.





**Table 2.** Salient feature of selected study catchments.

| station | river | GRDCno | latitude [°N] | longitude [°E] | drainage Area [km²] | mean annual discharge [m³s⁻¹] | start year | length (year) |
|---|---|---|---|---|---|---|---|---|
| Orsova, RO | Danube | 6742200 | 44.7 | 22.42 | 576232 | 5602 | 1840 | 151 |
| Decin, CZ | Elbe | 6140400 | 50.79 | 14.23 | 51123 | 309 | 1851 | 150 |
| Dresden, DE | Elbe | 6340120 | 51.05 | 13.73 | 53096 | 332 | 1806 | 208 |
| Elverum, NO | Gloma | 6731401 | 60.88 | 11.56 | 15426 | 251 | 1871 | 44 |
| Vargoens KRV, SW | Gota Alv | 6229500 | 58.35 | 12.37 | 46885.5 | 531 | 1807 | 212 |
| Wasserburg, DE | Inn | 6343100 | 48.05 | 12.23 | 11983 | 354 | 1827 | 177 |
| Muroleenkoski,FI | Kokemaenjoki | 6854104 | 61.85 | 23.910 | 6102 | 53.1 | 1863 | 155 |
| Blois, FR | Loire | 6123300 | 47.58 | -0.86 | 38240 | 362 | 1863 | 117 |
| Montjean, FR | Loire | 6123100 | 47.58 | 1.33 | 110000 | 911 | 1863 | 117 |
| Schweinfurt-Neuer Hafen | Main | 6335301 | 50.03 | 10.22 | 12715 | 103 | 1845 | 156 |
| Weurzburg, DE | Main | 6335500 | 49.79 | 9.92 | 14031 | 108 | 1824 | 177 |
| Smalininkai, LT | Nemunas | 6574150 | 55.07 | 22.57 | 81200 | 531 | 1812 | 185 |
| Basel Rheinhalle, CH | Rhine | 6935051 | 47.55 | 7.61 | 35897 | 1043 | 1869 | 140 |
| Basel Schifflaende, CH | Rhine | 6935052 | 47.55 | 7.58 | 35905 | 1042 | 1869 | 127 |
| Köln, DE | Rhine | 6335060 | 50.93 | 6.96 | 144232 | 2085 | 1817 | 195 |
| Rees, DE | Rhine | 6335020 | 51.75 | 6.39 | 159300 | 2251 | 1815 | 183 |
| Burgausen, DE | Salzach | 6343500 | 48.15 | 12.83 | 6649 | 258 | 1827 | 174 |
| Hann-Münden DE | Wesser | 6337400 | 51.42 | 9.64 | 12442 | 109 | 1831 | 182 |
| Bodenwerder, DE | Wesser | 6337514 | 51.97 | 9.51 | 15924 | 145 | 1839 | 175 |
| Vlotho DE | Wesser | 6337100 | 52.17 | 8.86 | 17618 | 170 | 1820 | 194 |
| Intschede, DE | Wesser | 6337200 | 52.96 | 9.12 | 37720 | 320 | 1857 | 154 |





## 3 Methods

This section is divided into three parts. The first part describes the selection and pre-processing of the reconstructed forcing (i.e., precipitation and temperature) for validation across Europe and the preparation of data for runoff simulation in several catchments. The hydrologic and data-driven models used, for runoff simulation are introduced in the second part. Finally, we

describe the methods for the evaluation of simulated runoff (including drought identification).

### 3.1 Data pre-processing

We prepared two datasets. The first consists of reconstructed forcings and the corresponding GHCN data for all available European stations with long records (see Section 2.3). We considered the selected GHCN stations and data from the corresponding grid cells of the reconstructed forcings for this forcing validation exercise. To understand how the reconstructed forcings match

the GHCN data across time scales, we aggregated both the reconstructed forcings and the GHCN data from seasons to 1, 2, . . . 30 years and calculated various goodness-of-fit (GOF) metrics (see further in Appendix A1).

The second dataset represents the data of 21 selected catchments and consists of reconstructed forcings and the proxy data and runoff for the calibration and validation of individual catchments (Fig. 2). The catchment average precipitation and temperature were estimated from the reconstructed forcings by averaging the grid cells covering the specific catchment boundary.

Similarly, we calculated the average catchment PDSI from the OWDA, and also selected the raw proxy data from inside the catchment or within a 100 km buffer around the catchment.

### 3.2 Hydrologic model (GR1A)

To simulate runoff in each catchment, we applied the annual time-scale hydrologic model, GR1A (Mouelhi et al., 2006). This model builds upon the work of Manabe (1969), considering dynamic storage and antecedent precipitation conditions. The

model consists of a simple mathematical equation with a single (optimized) parameter:

$$Q_i = P_i \left\{ 1 - \frac{1}{\left[ 1 + \left( \frac{0.8P_i + 0.2P_{i-1}}{XE_i} \right)^2 \right]^{0.5}} \right\} \tag{1}$$

where $Q$, $E$ and $P$ represent annual runoff, potential evapotranspiration and precipitation, respectively; $i$ denotes the year specific index. The parameter $X$ is optimized, maximizing the Nash-Sutcliffe efficiency (NSE) between the observed and modelled runoff data. The potential evapotranspiration was calculated using the temperature-based formula, provided by Oudin

et al. (2005).

### 3.3 Data-driven models

Data-driven methods, Artificial Neural Networks (ANNs; Kwak et al., 2020; Hu et al., 2018; Senthil Kumar et al., 2005) in particular, have been widely used for rainfall-runoff prediction. ANN algorithms are very flexible in describing non-linear





relations. The ANNs consist of artificial neurons organized in layers and connections that route the signal through the network.
Each connection has an associated weight that is optimized within the calibration (in the context of ANNs, known as training).
There are many kinds of ANN which differ in terms of structure and type of connections, as well as direction, functional forms
used for neuron activation or training.

In the present study, we considered two approaches : long short term memory (LSTM) neural networks and Bayesian
regularized neural networks (BRNN). These techniques are commonly used to determine the relationship between rainfall and
runoff (Hu et al., 2018; Xiang et al., 2020; Kratzert et al., 2018; Ye et al., 2021). We considered combinations of gridded forcing,
OWDA-based scPDSI, proxies and lagged gridded forcing as an input into the network for both model types. Specifically, the
network using only gridded forcing is referred to as "Gridded", the network with a combination of gridded forcing and natural
proxies is known as "Gridded+Proxies", the network with gridded forcing and OWDA scPDSI is termed as "Gridded+PDSI"
and finally the network which include lagged gridded forcing is referred to as "Gridded+Lag".

Figure A1 shows the architecture of LSTM, which is a modified version of the recurrent neural network, based on the back-
propagation algorithm (Hochreiter and Schmidhuber, 1997). In this structure, LSTM allows to learn a long-term data set and
controls the overfitting problem (Chen et al., 2020). LSTM generally consists of two unit states (hidden and cell states) and
three distinct gates (hidden, input and output). In this process, a given cell state saves the long-term memory at the previous
unit, while hidden states act as a working memory to process information inside the gates. These gates can determine which
information needs to be processed, remembered and transferred in the next state. With LSTM, different activation functions,
such as hyperbolic tangent $tanh$ and sigmoid $\sigma$, can be used to update unit states. The implementation of the LSTM is carried
out by means of R packages: "keras" (Arnold, 2017) and "tensorflow" (Abadi et al., 2016).

The training process of the LSTM is time consuming due to its inherent complexity. Therefore, the BRNN method was
proposed because of its fast learning and high convergence approximation. Moreover, the BRNN helps to tackle the complex
relationship between rainfall and runoff responses (Ye et al., 2021). This method implements the initial values of the ANN
parameters, using Bayesian regularization (Okut, 2016). Initial weights are set up as a prior distribution function during model
training, typically taken as a normal distribution. By applying Bayesian formulation, weight parameters keep updating prior
probability distribution to the posterior probability distribution. We trained this model in R using the "brnn" function of the
"caret" package (Kuhn, 2015).

In both cases, the model optimization runs were conducted several times, and the one with the best performance was consid-
ered for further evaluation. To reduce the likelihood of overfitting during the calibration/training, a fraction of the calibration
data was used to check the performance of an independent (or so-called "testing") set. In addition, the network parameters
(such as the number of neurons, activation functions, etc.) were iteratively tuned to yield fast convergence and good skill.

### 3.4 Goodness-of-fit assessment

We used a set of seven statistical metrics to assess the performance of simulated runoff, namely: Nash–Sutcliffe efficiency
(NSE), index of agreement (D), Pearson correlation (R), relative error in standard deviation (rSD), Kling-Gupta efficiency

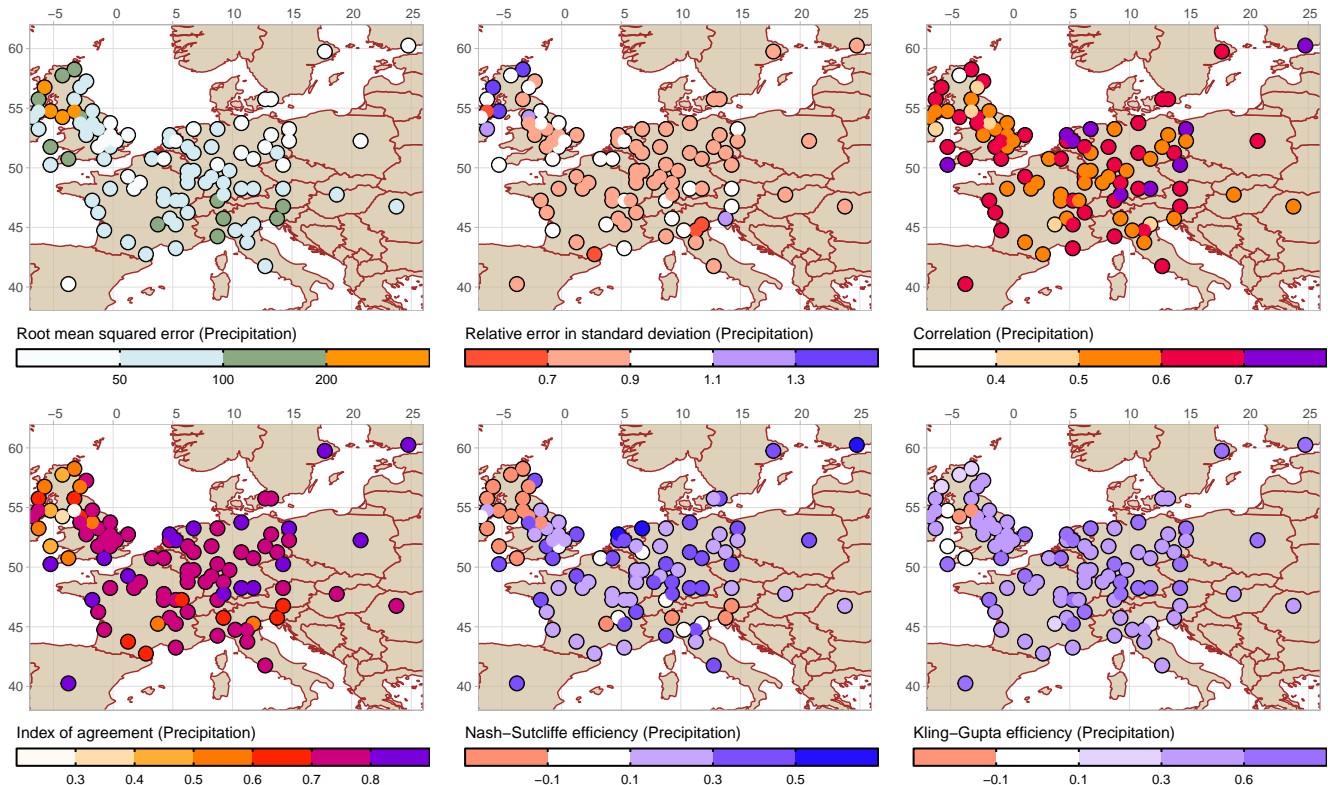

**Figure 2.** Validation of reconstructed precipitation (Pauling et al., 2006) against GHCN observations.

(KGE), root mean square error (RMSE), mean absolute error (MAE). The mathematical formulations of these metrics are provided in Appendix A1.

## 3.5 Runoff drought identification

To check the utility of our reconstruction, we finally explore how well the runoff droughts are represented in the simulations. Our study considers hydrological droughts, defined based on the streamflow deficit, following the threshold level approach (Yevjevich, 1967; Rivera et al., 2017; Sung and Chung, 2014). This approach is typically used for daily or monthly time scales, considering 0.1 or 0.2 quantile threshold levels. To accommodate the annual scale as used here, we defined the start of the drought, when the runoff anomaly falls below the 0.33 quantile (regular drought) and the 0.05 quantile (extreme drought).

The drought persists until the runoff rises above the threshold again. Drought length and severity (the cumulative difference of runoff and the threshold) were then calculated for each identified drought year. Hydrologic drought series can be further assessed to understand the critical aspects of runoff (temporal) dynamics and to classify past droughts in Europe (Cook et al., 2015; Wetter and Pfister, 2013).



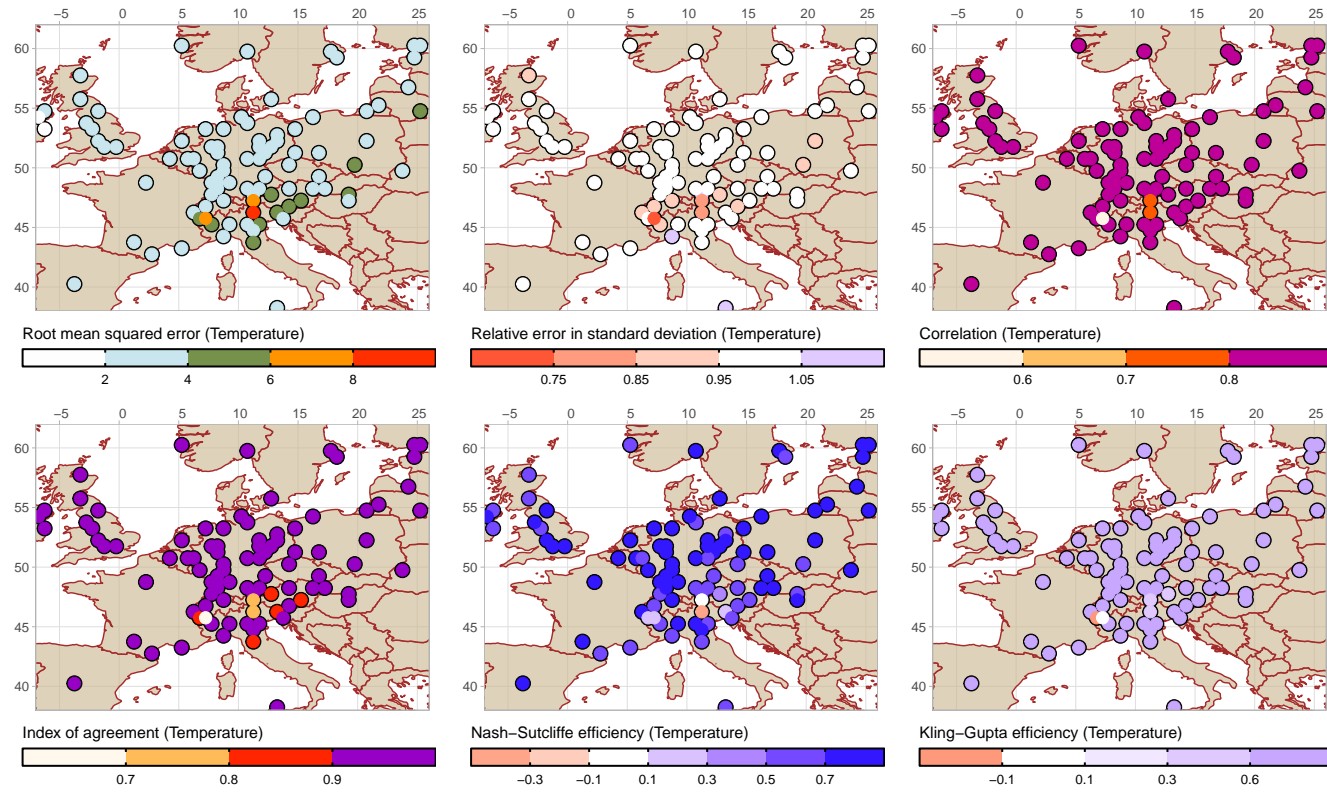

**Figure 3.** Validation of reconstructed temperature (Luterbacher et al., 2004) against GHCN observations.

## 4    Results and discussion

In this section, we analyze the 500-year-long reconstruction over space and time across Europe. Firstly, we provide a comparison between the GHCN observed precipitation and temperature, and the corresponding grid cells from Pauling et al. (2006) and Luterbacher et al. (2004) reconstructions. Next, the reconstructed runoff series for the selected catchments are evaluated against the corresponding observed GRDC runoff data.

Two distinct model types were investigated, i.e., a process-based hydrological model (GR1A) and two data-driven models (BRNN and LSTM). While the former takes gridded precipitation and temperature as an input, in the case of the latter, we also considered PDSI, natural proxies and lagged reconstructed precipitation and temperature fields, as shown in Tab. 4. Statistical metrics, such as NSE, KGE, RMSE, MAE, R and D (Appendix A1) are used to quantify the predictive skills of the models examined.



## 4.1 Evaluation of reconstructed precipitation and temperature fields

The 500-year long paleoclimate reconstructions of precipitation (P) and temperature (T) were validated against the GHCN observation data. The spatial map for the comparison is given in Figs. 2 and 3. The reconstructed data are verified against observational P and T across 99 and 94 European sites, respectively. Figure 2 shows that the correlation coefficient (R) of P reconstruction at most of the sites is above 0.5; the index of agreement (D) is larger than 0.6; KGE and NSE are showing values below 0.5 (NSE) and 0.6 (KGE); the rSD measurement is greater than 0.7 and RMSE varies between 50 and 100. We

found relatively good performance values for temperature reconstruction, as depicted in Fig. 3. In this case, RMSE, estimated between reconstructed and observational T, is around $0.2°C$; rSD fluctuates between 0.95 and 1.05, while R is higher than 0.84 and D is above 0.90. The NSE and KGE values were above 0.5 at many stations. Some stations indicated a worse performance and could not adequately capture the observed temperature variability.

Furthermore, we tested the skill of gridded reconstructed forcings to capture the multi-temporal characteristics of observed P

and T dynamics, i.e., aggregated time-scale features ranging from seasonal to 30-year data. To this end, the seasonal values of the P and T data series were aggregated from 0.25 to a 30-year period (with annual increments) with no overlapping windows. The GOF statistics (Section 3.4) between each GHCN station and the corresponding reconstruction grid cell were estimated. In Figure 4, we present the median gof statistics (black line), the ranges between the $25^{th}$ and $75^{th}$ (light envelope) and the $10^{th}$ and $90^{th}$ quantiles (dark envelope) of the distribution of the gof statistics over the stations for each aggregated time-step.

The RMSE for precipitation and temperature drops from initially high values for seasonal scales to relatively stable values for aggregations with a duration greater than 10 years. This is expected since the RMSE depends on the number of observations. With regards to other statistics, except for correlation which shows relatively stable values over aggregations, it is evident that the reconstruction skill decreases the greater the (aggregation) time-scale. In particular, the variance is underestimated and this underestimation is more substantial for long aggregations (see rSD panel in Fig. 4). This may imply that the utility of multi-year

(drought) assessment, utilizing the reconstructed forcing datasets can be limited (and should be interpreted with caution).

It is worth noting that the large spread of gof statistics is mainly due to the outlying values at the grid cell, located along the boundary of the domain (i.e., the interface between land and sea/ocean) and high elevations (cf. also Figs. 2 and 3). In general, reconstructed precipitation exhibits greater differences from observations than temperature. This may be because the proxies considered in the reconstruction rely on different seasons and climate conditions. Additionally, the shortest available

instrumental data before the $20^{th}$ century could encounter certain technical errors, such as problems with instrumental tools, station relocation and dating issues (Dobrovolný et al., 2010). Moreover, other studies (e.g., Ljungqvist et al., 2020) stated that the precipitation series employed for the reconstructions were relatively shorter and more erroneous than the temperature series before the $20^{th}$ century (Pauling et al., 2006; Harris et al., 2014). Finally, the chosen statistical technique (principal component regression) could also contribute to variance inflation with larger time-scales (Pauling et al., 2006).



Earth System
Science
Data

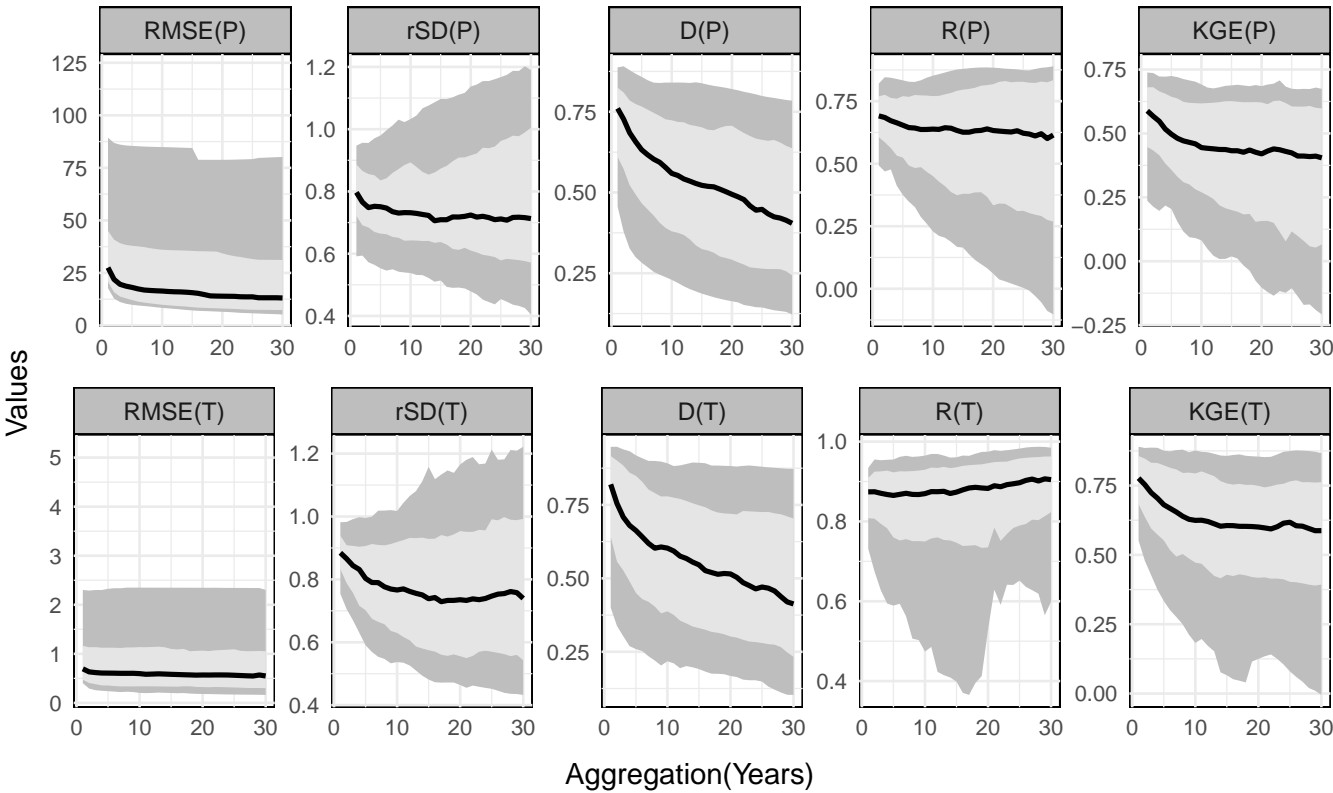

**Figure 4.** 30 year rolling window of the 500 years gridded data and GRDC observations across 30°N–45°N, 10°W–40°E, the envelope span of two quantiles between the (10th and 90th), (10,75) percentiles of grid cell values for each region. The median value is represented with black thick line, while vertical axis shows the corresponding metric scale separately, for precipitation and temperature

## 4.2 Assessment of the reconstructed runoff simulations

For runoff prediction, we have considered several input variables for the models (i.e., the GR1A hydrologic model, the BRNN and LSTM data-driven models), as detailed in Table 4. The available GRDC observed runoff time-series at each gauging location were split into two parts: calibration (1900–2000), used to identify model parameters and validation (prior to 1900), for independent verification using the GOF statistics.

The GR1A conceptual hydrological model was driven by the gridded reconstruction of P and T to simulate the runoff for each catchment separately. The simulated runoff series were then compared to the corresponding GRDC observations and the results were summarized by means of GOF statistics. As can be seen in Table 3, the correlation and NSE statistics for calibration achieve reasonable results at most of the catchments, with a few exceptions (i.e., Kokemenjoki, Goeta, Nemunas and Inn). These (relatively poorer catchment skills in northern Europe) are in line with the previous findings of Seiller et al. (2012) who noted that the lumped hydrological models often exhibit larger uncertainties and fail to capture the extreme catchment values (both high and low). Another study of Fathi et al. (2019) suggested that the performance of the GR1A model is less





efficient than the new Budyko framework SARIMA model in simulating the annual runoff across the Blue Nile and Danube catchment. This is because the simplified nature of the model does not easily capture the complex relationship between rainfall and runoff variability.

The NSE and R statistics (Tab. 3) for the BRNN models indicate a significant improvement in runoff prediction, as compared to the results obtained through the GR1A model. This is especially true with regard to the catchment in Switzerland (Basel Rheinhalle NSE increases from 0.27 to 0.73 for calibration, and 0.2 to 0.54 for validation). Inclusion of climate-related natural proxies in addition to the reconstructed forcings as an input to the model [BRNN(Gridded+Proxies)] did not make any significant contribution to the model skill. However, the combination of scPDSI from OWDA with reconstructed forcing [BRNN(Gridded+PDSI)], greatly increased the performance (NSE from 0.2 to 0.62).

Similarly at most of the sites, the simulation based on reconstructed forcings in combination with OWDA scPDSI, yielded a positive correlation with the observed runoff. Other metrics (RMSE, MAE, KGE and D) are shown in Tabs. S3 and S4 in Supplementary material). Across many study locations, the combination of reconstructed forcings with their (one year) lagged version performed the best in terms of rapid convergence (the number of iterations needed) and high accuracy from all input combinations (Gridded, Gridded+PDSI, Gridded+Proxies and Gridded+Lag) for both models (BRNN, LSTM). In general, statistical values in heat-maps indicate that the neural network algorithms are more skilled for runoff prediction than the GR1A model. For the validation period, the mean NSE (across all catchments) for the GR1A model is 0.1638, for the BRNN(Gridded+Lag) it is 0.6836 and improves to 0.7347 for the LSTM(Gridded+Lag). In the case of the mean KGE, GR1A is 0.617, BRNN(gridded+lag) is 0.737 and LSTM(gridded+lag) is 0.785.

To further demonstrate the differences between the individual models, we show the simulated runoff series for all models for those catchments with the highest (Blois, Loire) and lowest (Smalininkai, Nemunas) performance in Figure 5. The performance of the models is comparable during the calibration period for the Loire River. Clearly, all data-driven models are capable of mimicking the observed runoff, while the GR1A model exhibited certain minor deviations, primarily until 1930. In the validation period, the differences between the models are more visible, in particular, for above-average flows. At the beginning of the validation period (1870–1880) all models failed to simulate the high annual flows.

In the case of Nemaunas catchment, the GR1A simulation deviates extremely from the observed data and cannot capture the mean flow level. However, the calibration is poor even for the data-driven models and, does not simulate the year-to-year variability appropriately. Interestingly, the error in the GR1A model is less in relation to validation than calibration. The data-driven models perform in a similar way to that of calibration, with only minor differences between the two periods. Looking at the gof statistics, the models considering OWDA-based scPDSI or lagged forcings (e.g., $P_{t-1}$) perform slightly better in terms of KGE than the other model configurations. This improvement may be due to a better representation of temporal dependency structure, introduced either by scPDSI or a consideration of the forcing values from the previous year in the case of LSTM(Gridded+Lag) and BRNN(Gridded+Lag).





### Correlation calibration

| | GR1A_Model(Gridded) | BRNN(Gridded) | LSTM(Gridded) | BRNN(Gridded+Proxies) | LSTM(Gridded+Proxies) | BRNN(Gridded+PDSI) | LSTM(Gridded+PDSI) | BRNN(Gridded+Lag) | LSTM(Gridded+Lag) |
|---|---|---|---|---|---|---|---|---|---|
| Orsova-Danube | 0.74 | 0.84 | 0.86 | 0.89 | 0.91 | 0.87 | 0.89 | 0.89 | 0.93 |
| Decin-Elbe | 0.62 | 0.8 | 0.82 | 0.82 | 0.86 | 0.83 | 0.85 | 0.86 | 0.87 |
| Dresden-Elbe | 0.65 | 0.8 | 0.81 | 0.81 | 0.84 | 0.82 | 0.77 | 0.84 | 0.85 |
| Elverum-Glama | 0.63 | 0.72 | 0.79 | 0.76 | 0.74 | 0.75 | 0.78 | 0.74 | 0.81 |
| Vargoens KRV- Goeta | 0.36 | 0.45 | 0.59 | 0.5 | 0.46 | 0.49 | 0.55 | 0.76 | 0.78 |
| Wasserburg-Inn | 0.65 | 0.77 | 0.79 | 0.78 | 0.86 | 0.77 | 0.76 | 0.78 | 0.88 |
| Muroleekoski-Kokemenjoki | 0.47 | 0.79 | 0.53 | 0.79 | 0.81 | 0.82 | 0.81 | 0.86 | 0.84 |
| Blois-Loire | 0.82 | 0.84 | 0.87 | 0.86 | 0.89 | 0.84 | 0.9 | 0.88 | 0.9 |
| Montjean-Loire | 0.81 | 0.86 | 0.89 | 0.87 | 0.83 | 0.86 | 0.9 | 0.91 | 0.91 |
| NeuerHafen-Main | 0.7 | 0.71 | 0.74 | 0.74 | 0.77 | 0.74 | 0.76 | 0.78 | 0.8 |
| Wuerzburg-Main | 0.66 | 0.67 | 0.73 | 0.71 | 0.66 | 0.71 | 0.77 | 0.75 | 0.84 |
| Smalininkai-Nemunas | 0.29 | 0.51 | 0.53 | 0.52 | 0.52 | 0.55 | 0.54 | 0.6 | 0.65 |
| BaselRheinhalle-Rhine | 0.71 | 0.86 | 0.87 | 0.89 | 0.89 | 0.87 | 0.87 | 0.86 | 0.91 |
| Baselschifflaende-Rhine | 0.72 | 0.87 | 0.9 | 0.9 | 0.9 | 0.88 | 0.87 | 0.88 | 0.92 |
| Koeln-Rhine | 0.86 | 0.86 | 0.87 | 0.89 | 0.84 | 0.87 | 0.89 | 0.9 | 0.94 |
| Rees-Rhine | 0.82 | 0.86 | 0.89 | 0.89 | 0.91 | 0.87 | 0.89 | 0.9 | 0.92 |
| Burghausen-Salzach | 0.63 | 0.64 | 0.69 | 0.66 | 0.72 | 0.74 | 0.83 | 0.64 | 0.76 |
| Hann-Munden-Wesser | 0.81 | 0.8 | 0.82 | 0.82 | 0.84 | 0.81 | 0.82 | 0.86 | 0.9 |
| Bodenwerder-Wesser | 0.81 | 0.81 | 0.83 | 0.82 | 0.81 | 0.81 | 0.83 | 0.86 | 0.93 |
| Vlotho-Wesser | 0.75 | 0.81 | 0.83 | 0.82 | 0.71 | 0.81 | 0.84 | 0.87 | 0.85 |
| Intschede-Wesser | 0.75 | 0.78 | 0.82 | 0.8 | 0.83 | 0.78 | 0.8 | 0.85 | 0.84 |

### Correlation validation

| | GR1A_Model(Gridded) | BRNN(Gridded) | LSTM(Gridded) | BRNN(Gridded+Proxies) | LSTM(Gridded+Proxies) | BRNN(Gridded+PDSI) | LSTM(Gridded+PDSI) | BRNN(Gridded+Lag) | LSTM(Gridded+Lag) |
|---|---|---|---|---|---|---|---|---|---|
| Orsova-Danube | 0.64 | 0.74 | 0.73 | 0.56 | 0.53 | 0.76 | 0.76 | 0.75 | 0.73 |
| Decin-Elbe | 0.62 | 0.65 | 0.61 | 0.59 | 0.51 | 0.65 | 0.67 | 0.7 | 0.71 |
| Dresden-Elbe | 0.5 | 0.66 | 0.65 | 0.65 | 0.57 | 0.67 | 0.68 | 0.71 | 0.7 |
| Elverum-Glama | 0.32 | 0.63 | 0.66 | 0.09 | 0.5 | 0.51 | 0.57 | 0.62 | 0.49 |
| Vargoens KRV- Goeta | 0.32 | 0.32 | 0.4 | 0.23 | -0.08 | 0.43 | 0.33 | 0.47 | 0.49 |
| Wasserburg-Inn | 0.69 | 0.73 | 0.71 | 0.73 | 0.66 | 0.72 | 0.69 | 0.72 | 0.71 |
| Muroleekoski-Kokemenjoki | 0.4 | 0.53 | 0.4 | 0.53 | 0.52 | 0.54 | 0.53 | 0.62 | 0.62 |
| Blois-Loire | 0.74 | 0.82 | 0.81 | 0.69 | 0.65 | 0.82 | 0.8 | 0.8 | 0.79 |
| Montjean-Loire | 0.74 | 0.73 | 0.68 | 0.72 | 0.68 | 0.75 | 0.73 | 0.79 | 0.74 |
| NeuerHafen-Main | 0.62 | 0.77 | 0.72 | 0.7 | 0.71 | 0.72 | 0.64 | 0.79 | 0.79 |
| Wuerzburg-Main | 0.7 | 0.75 | 0.6 | 0.65 | 0.47 | 0.77 | 0.66 | 0.77 | 0.73 |
| Smalininkai-Nemunas | 0.37 | 0.42 | 0.42 | 0.34 | 0.24 | 0.43 | 0.38 | 0.44 | 0.44 |
| BaselRheinhalle-Rhine | 0.83 | 0.83 | 0.78 | 0.8 | 0.71 | 0.84 | 0.83 | 0.85 | 0.8 |
| Baselschifflaende-Rhine | 0.83 | 0.83 | 0.78 | 0.78 | 0.78 | 0.84 | 0.83 | 0.84 | 0.82 |
| Koeln-Rhine | 0.81 | 0.86 | 0.85 | 0.82 | 0.44 | 0.86 | 0.84 | 0.88 | 0.86 |
| Rees-Rhine | 0.78 | 0.83 | 0.81 | 0.81 | 0.7 | 0.8 | 0.79 | 0.82 | 0.8 |
| Burghausen-Salzach | 0.37 | 0.67 | 0.64 | 0.65 | 0.47 | 0.49 | 0.37 | 0.68 | 0.67 |
| Hann-Munden-Wesser | 0.62 | 0.78 | 0.74 | 0.55 | 0.55 | 0.77 | 0.69 | 0.82 | 0.77 |
| Bodenwerder-Wesser | 0.65 | 0.8 | 0.77 | 0.72 | 0.62 | 0.8 | 0.75 | 0.85 | 0.8 |
| Vlotho-Wesser | 0.4 | 0.74 | 0.73 | 0.55 | 0.61 | 0.73 | 0.72 | 0.78 | 0.76 |
| Intschede-Wesser | 0.63 | 0.74 | 0.74 | 0.75 | 0.7 | 0.74 | 0.75 | 0.82 | 0.82 |

### NSE calibration

| | GR1A_Model(Gridded) | BRNN(Gridded) | LSTM(Gridded) | BRNN(Gridded+Proxies) | LSTM(Gridded+Proxies) | BRNN(Gridded+PDSI) | LSTM(Gridded+PDSI) | BRNN(Gridded+Lag) | LSTM(Gridded+Lag) |
|---|---|---|---|---|---|---|---|---|---|
| Orsova-Danube | 0.25 | 0.71 | 0.74 | 0.79 | 0.82 | 0.75 | 0.78 | 0.8 | 0.86 |
| Decin-Elbe | 0.19 | 0.64 | 0.68 | 0.67 | 0.73 | 0.69 | 0.72 | 0.74 | 0.76 |
| Dresden-Elbe | 0.3 | 0.65 | 0.65 | 0.66 | 0.71 | 0.67 | 0.6 | 0.7 | 0.71 |
| Elverum-Glama | 0.28 | 0.51 | 0.62 | 0.57 | 0.53 | 0.56 | 0.57 | 0.54 | 0.66 |
| Vargoens KRV- Goeta | -1.16 | 0.2 | 0.34 | 0.24 | 0.11 | 0.24 | 0.29 | 0.58 | 0.58 |
| Wasserburg-Inn | -0.31 | 0.59 | 0.62 | 0.61 | 0.73 | 0.6 | 0.58 | 0.61 | 0.76 |
| Muroleekoski-Kokemenjoki | -0.29 | 0.63 | 0.27 | 0.63 | 0.63 | 0.68 | 0.65 | 0.73 | 0.7 |
| Blois-Loire | 0.66 | 0.71 | 0.75 | 0.75 | 0.77 | 0.71 | 0.8 | 0.77 | 0.79 |
| Montjean-Loire | 0.65 | 0.74 | 0.79 | 0.75 | 0.68 | 0.73 | 0.81 | 0.82 | 0.82 |
| NeuerHafen-Main | 0.44 | 0.51 | 0.54 | 0.54 | 0.58 | 0.55 | 0.57 | 0.6 | 0.64 |
| Wuerzburg-Main | 0.35 | 0.45 | 0.52 | 0.5 | 0.42 | 0.5 | 0.58 | 0.56 | 0.71 |
| Smalininkai-Nemunas | -2.35 | 0.26 | 0.26 | 0.27 | 0.27 | 0.3 | 0.29 | 0.36 | 0.41 |
| BaselRheinhalle-Rhine | 0.27 | 0.73 | 0.75 | 0.78 | 0.78 | 0.76 | 0.76 | 0.75 | 0.82 |
| Baselschifflaende-Rhine | 0.27 | 0.76 | 0.79 | 0.81 | 0.8 | 0.78 | 0.74 | 0.78 | 0.84 |
| Koeln-Rhine | 0.69 | 0.74 | 0.75 | 0.79 | 0.71 | 0.75 | 0.78 | 0.82 | 0.87 |
| Rees-Rhine | 0.59 | 0.74 | 0.78 | 0.79 | 0.82 | 0.76 | 0.79 | 0.81 | 0.86 |
| Burghausen-Salzach | 0.33 | 0.41 | 0.48 | 0.44 | 0.52 | 0.55 | 0.68 | 0.41 | 0.57 |
| Hann-Munden-Wesser | 0.62 | 0.64 | 0.65 | 0.67 | 0.69 | 0.65 | 0.67 | 0.74 | 0.8 |
| Bodenwerder-Wesser | 0.63 | 0.65 | 0.69 | 0.67 | 0.65 | 0.65 | 0.67 | 0.75 | 0.86 |
| Vlotho-Wesser | 0.51 | 0.65 | 0.66 | 0.67 | 0.41 | 0.65 | 0.68 | 0.77 | 0.72 |
| Intschede-Wesser | 0.52 | 0.6 | 0.65 | 0.63 | 0.68 | 0.6 | 0.6 | 0.73 | 0.69 |

### NSE validation

| | GR1A_Model(Gridded) | BRNN(Gridded) | LSTM(Gridded) | BRNN(Gridded+Proxies) | LSTM(Gridded+Proxies) | BRNN(Gridded+PDSI) | LSTM(Gridded+PDSI) | BRNN(Gridded+Lag) | LSTM(Gridded+Lag) |
|---|---|---|---|---|---|---|---|---|---|
| Orsova-Danube | -2.37 | 0.51 | 0.5 | -0.09 | 0.28 | 0.57 | 0.58 | 0.48 | 0.39 |
| Decin-Elbe | -1.42 | 0.4 | 0.36 | 0.34 | 0.18 | 0.4 | 0.44 | 0.43 | 0.47 |
| Dresden-Elbe | -0.4 | 0.42 | 0.4 | 0.39 | 0.3 | 0.4 | 0.41 | 0.51 | 0.48 |
| Elverum-Glama | 0.04 | 0.11 | 0.14 | -1.29 | 0.06 | 0.01 | 0.19 | 0.1 | -0.28 |
| Vargoens KRV- Goeta | -0.77 | -0.17 | -0.9 | -0.28 | -0.18 | -0.19 | -0.28 | -0.13 | 0.06 |
| Wasserburg-Inn | -0.96 | 0.52 | 0.49 | 0.52 | 0.4 | 0.51 | 0.47 | 0.52 | 0.45 |
| Muroleekoski-Kokemenjoki | -0.89 | 0.26 | 0.01 | 0.26 | 0.2 | 0.29 | 0.27 | 0.37 | 0.36 |
| Blois-Loire | 0.48 | 0.67 | 0.65 | 0.44 | 0.42 | 0.67 | 0.64 | 0.62 | 0.58 |
| Montjean-Loire | 0.28 | 0.46 | 0.38 | 0.4 | 0.44 | 0.48 | 0.42 | 0.55 | 0.48 |
| NeuerHafen-Main | 0.01 | 0.48 | 0.45 | 0.38 | 0.45 | 0.46 | 0.39 | 0.56 | 0.6 |
| Wuerzburg-Main | -0.58 | 0.46 | 0.25 | 0.4 | 0.15 | 0.57 | 0.38 | 0.51 | 0.37 |
| Smalininkai-Nemunas | -1.28 | 0.1 | 0.1 | -0.01 | -0.04 | 0.16 | 0.08 | 0.02 | 0.09 |
| BaselRheinhalle-Rhine | 0.2 | 0.54 | 0.52 | 0.54 | 0.4 | 0.57 | 0.59 | 0.6 | 0.54 |
| Baselschifflaende-Rhine | 0.23 | 0.53 | 0.52 | 0.46 | 0.49 | 0.56 | 0.66 | 0.59 | 0.57 |
| Koeln-Rhine | 0.39 | 0.7 | 0.69 | 0.07 | 0.18 | 0.65 | 0.66 | 0.71 | 0.67 |
| Rees-Rhine | 0.5 | 0.65 | 0.64 | 0.52 | 0.48 | 0.61 | 0.58 | 0.64 | 0.62 |
| Burghausen-Salzach | -1.28 | 0.08 | 0.07 | 0.16 | 0.01 | -0.14 | -0.21 | 0.11 | -0.07 |
| Hann-Munden-Wesser | -0.13 | 0.55 | 0.54 | 0.16 | 0.17 | 0.53 | 0.46 | 0.64 | 0.52 |
| Bodenwerder-Wesser | 0.23 | 0.51 | 0.55 | 0.36 | 0.37 | 0.51 | 0.5 | 0.58 | 0.35 |
| Vlotho-Wesser | -0.21 | 0.25 | 0.36 | -0.24 | 0.37 | 0.21 | 0.35 | 0.32 | 0.36 |
| Intschede-Wesser | 0.34 | 0.39 | 0.48 | 0.51 | 0.48 | 0.4 | 0.51 | 0.48 | 0.53 |

**Table 3.** The correlation coefficient (top) and NSE (bottom) for calibration (left) and validation (right) of the considered models for 21 study catchments. The y axis consists of water gauge stations (name and relevant river) in Central Europe, Alps and Lithuania. The rectangular black frames represent the catchments with satisfactory validation.



**Figure 5.** Comparison between the models for the station with the best (Bloise-Loire River, top) and the worst (Smalininkai-Nemaunas River, bottom) model fit.

### 4.3 The runoff reconstruction datasets

As a first step, we excluded the catchments that exhibited poor performance in validation (see Table 3). As a threshold, we considered validation NSE of 0.5 for at least one model, following the approach used by Ayzel et al. (2020). In this step, we excluded seven catchments (Vlotho-Wesser, Decin-Elbe, Burghausen-Salzach, Smalininkai-Nemaunas, Vargoens KRV- Goeta, Elverum-Glama, Muroleekoski-Kokemenjoki) out of 21, ending up with a set of simulations for 14 catchments (highlighted by the thick box in Tab. 3).

Secondly, we identified the best models for each of the 14 selected catchment, considering the gofs based on NSE and R greater than 0.5 and 0.70, respectively, for the validation period. In addition, the model performance with respect to the remaining measures (D, KGE, RMSE and MAE) was also considered. Eventually, we decided to utilize that model since the metrics


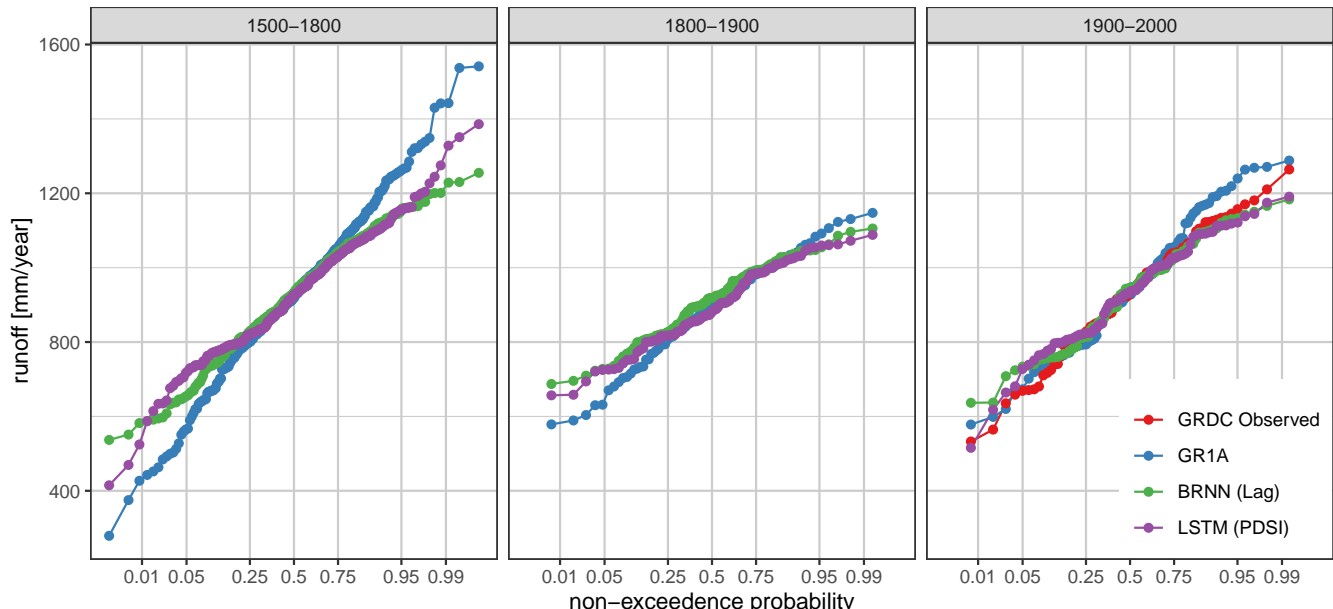

**Figure 6.** Distribution functions for BRNN(Lag), LSTM(PDSI), i.e. the best two models, GR1A and observed data (OBS) for the periods 1500–1800, 1800–1900 and 1900–2000 over Basel Rheinhalle-Rhine catchment. The values on the horizontal axis are transformed using the "probit" function.

used (NSE, KGE, R, D, RMSE, MAE) to produce better results in one particular model. The resulting selected models are shown in Table 4. The combination of gridded forcing with lagged values results in the best performance over nine catchments,

of which seven are driving the BRNN and the remainder the LSTM. The LSTM with gridded forcing and OWDA-scPDSI was best in one case, and the remaining four were most appropriately simulated with the BRNN and BRNN[Gridded+PDSI]. It should be noted that the differences between the models performing well are small, as noted in Figure 5 and further demonstrated in Figure 6. The latter figure compares the distribution functions of runoff for the periods 1500–1800, 1800–1900 and 1900–2000, as simulated by the BRNN(Lag) and LSTM (PDSI) – the two best performing models – and the GR1A (the most

distinctive simulations from the best model) with the distribution of the observed runoff for the Basel-Rheinhalle Rhine catchment. For the calibration period (post-1900), the models perform well except the GR1A, which generally overestimated the observed maxima. The BRNN and LSTM simulations are very similar for the validation period except for the top and bottom 5% in 1500–1800. The GR1A simulation showed significant differences for the entire observed distribution, thus overestimating/underestimating the maxima/minima. The difference from the best model can be expressed in terms of KGE – even here,

it was evident that the GR1A model deviated considerably (KGE 0.6–0.7) while the LSTM is very similar to the BRNN (KGE 0.92–0.96).

The resulting 14 runoff reconstructions are available at https://doi.org/10.6084/m9.figshare.15178107 and are shown in supplementary figures (Figs. S1, S2, and S3). As an additional validation for these series, we present the plots of the observed



runoff versus the reconstructed runoff in Fig. 7. The simulated series are generally consistent with the observed runoff, es-
pecially for the Montjean-Loire, Köln-Rhine, and Basel Schifflaende-Rhine catchments, which exhibit the best relationship
between the observed and the simulated runoff.

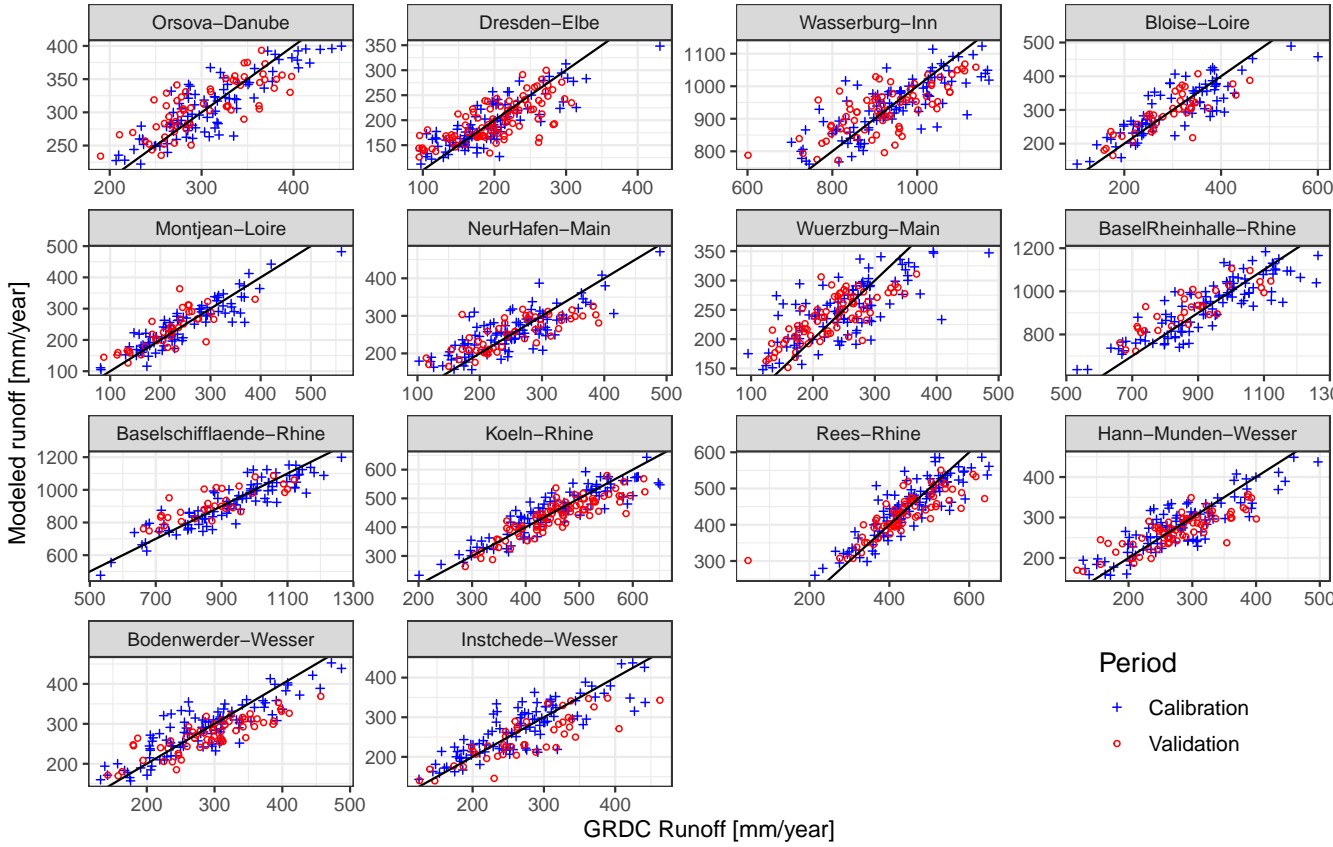

**Figure 7.** Observed and simulated runoff for 14 selected catchments in the calibration and validation periods

**Table 4.** Selection of best model for runoff in individual catchments

| Models | Catchments |
|---|---|
| BRNN(Gridded) | Blois-Loire, Rees-Rhine |
| BRNN(Gridded+PDSI) | Wuerzburg-Main and Orsova-Danube |
| BRNN(Gridded+Lag) | Montjean-Loire, Köln-Rhine, Hann-Munden-Wesser, Dresden-Elbe, BaselRheinhalle-Rhine, Bodenwerder-Wesser, Wasserburg-Inn |
| LSTM(Gridded+Lag) | NeuerHafen-Main, Intschede-Wesser |
| LSTM(Gridded+PDSI) | Baselschifflaende-Rhine |



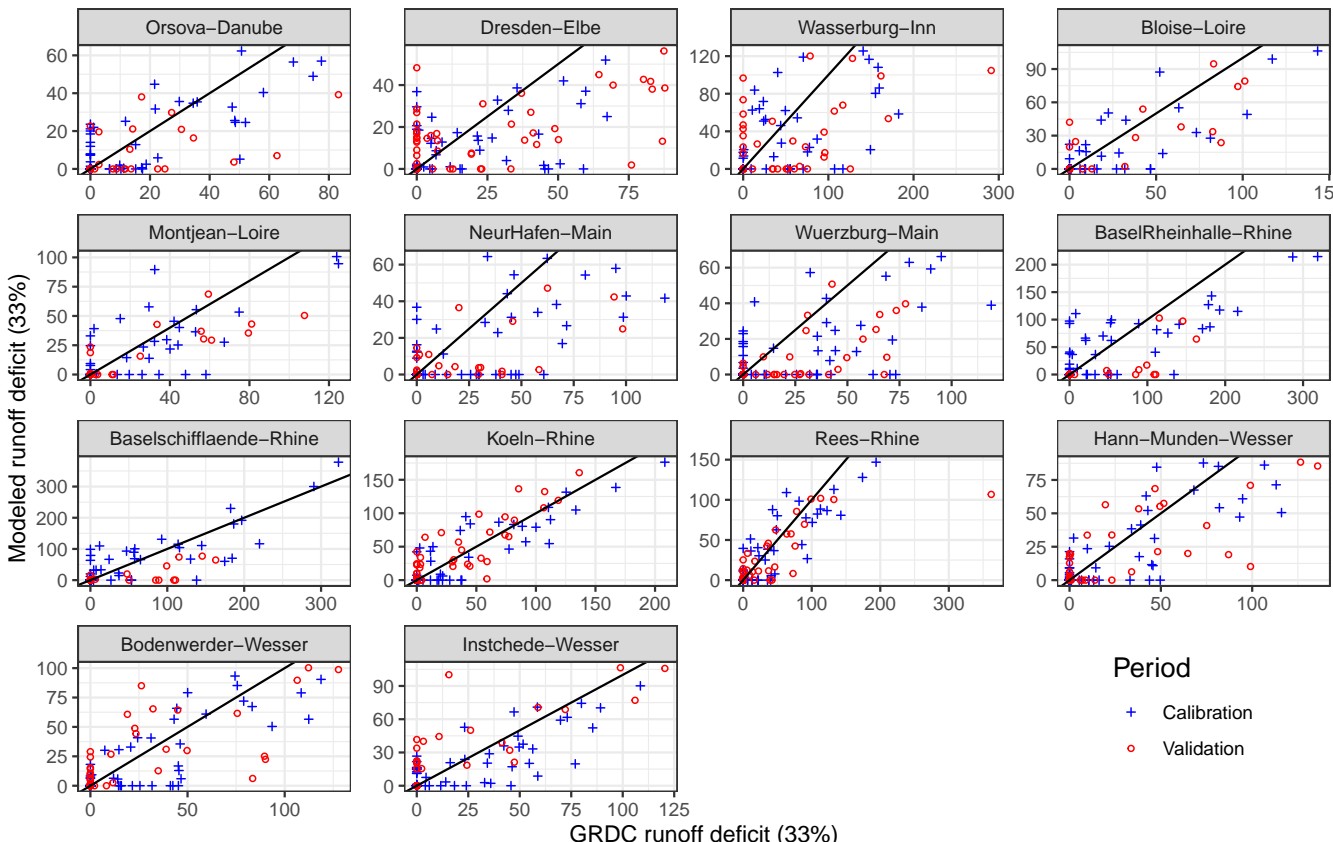

**Figure 8.** Observed and simulated runoff deficit of 14 selected catchments in the calibration and validation period.

### 4.4 Identification of low flows and significant hydrological drought events

In the final step of the analysis, we compared the droughts identified in the reconstructions with the GRDC observed series (Fig. 7). The match between the simulated and observed runoff deficit is less compared to the annual runoff time series. For most

of the stations, the simulated deficit is lower than the corresponding observed estimates. This suggests that the reconstructed precipitation and temperature fields do not represent the inter-annual variability correctly, which is in line with findings from Fig. 4. Despite a widespread issue with the representation of inter-annual persistence, Fig. 8 shows that the runoff deficits are simulated reasonably well for the Rees-Rhine and Köln-Rhine catchments.

In the next step, we contrasted reconstructed drought patterns over the last 500 years, with data available from documentary

evidence and other sources. In the case of extreme droughts, we considered the $q_{0.05}$ threshold before 2000 CE. Low flow analysis since 1500 and the maximum/minimum deficit values of catchments are shown in Tab. 5. In the $16^{th}$ century, the years 1536, 1540 and 1590 are associated with significant runoff deficits. The event of 1540, had already been reported (Brázdil et al., 2019; Cook et al., 2015; Brázdil et al., 2013) as the worst event of the $16^{th}$ century and was also more severe in terms of





hydrologic shifting. In 1540, almost 90% of the Rhine and Elbe River catchments (Basel and Cologne) experienced low yearly

discharge, which ranked as the greatest low flows in the last five centuries (Leggewie and Mauelshagen, 2018). The seasonal precipitation was also deficient and was evident primarily in Central Europe and England (Dobrovolný et al., 2010). Wetter and Pfister 2013 stated that the spring and summer of 1540 was likely to have been warmer than the comparable period during the 2003 drought. The simulation shows that the drought during 1540 was evident in most study catchments, such as the Rhine, Main, Wesser, Loire and Danube, except Wasserburg-INN. In the $17^{th}$ century, the years 1603, 1616, 1631, 1666, 1669, 1676,

1681, 1684 and 1686 were simulated as exceptionally low-flow years. Furthermore, two events (1686 and 1669) were associated with the maximum water deficit across several study catchments. Baselschifflaende-Rhine catchment is a good example of this, which experienced a severe runoff deficit during 1669. Alternatively, 26 remarkable droughts have been captured in the Köln-Rhine catchment over the past 500 years, and the year 1686 reached the highest runoff deficit (156 mm/year). In addition, 1616 was the driest year of the $17^{th}$ century, the so-called "drought of the century" (Brázdil et al., 2013), which significantly

impacted the major rivers in Europe (e.g., Rhine, Main and Wesser). Brázdil et al. (2018) identified three unusual drought periods (1540, 1616 and 1718–19) over the Czech lands, highlighting the 1616 drought, which caused widespread famine, dried up the Elbe river watershed and altered the climate of neighboring nations (Switzerland and Germany). The hunger stone of the Elbe River also revealed the exceptionally dry year of 1616 (Brázdil et al., 2013). During the $18^{th}$ century, a similar level of runoff deficit was simulated in the years 1706 and 1719.

During the $19^{th}$ century, the years 1863, 1864, 1874, 1893 and 1899, were recognized as drought years in all catchments, while in the $20^{th}$ century, the driest periods occurred in 1921, 1934, 1949 and 1976. The 1921 drought in the Blois-Loire, Rees-Rhine, Köln-Rhine, Orsova-Danube, BaselRheinhalle-Rhine and Baselschifflanede-Rhine catchments was ranked as the most exceptional drought in the $20^{th}$ century. Three catchments (BaselRheinhalle-Rhine, Baselschifflanede-Rhine and Blois-Loire) were simulated with a high deficit for the year 1921. A noticeable increase in temperature was experienced across Europe and

certain areas were notably affected by a heat wave in July of that year. The majority of Central Europe, southern England and Italy were affected by this drought, including London, where the rainfall was found to have decreased by 50 to 60% relative to the average (Cook et al., 2015). Similar to our results, certain photographs from the Dutch newspaper (De Telegraaf) show the lowest river flows in the Rhine (Switzerland), Molesey Weir (the Thames River, UK) and Loire River (France, van der Schrier et al., 2021). The precipitation totals were recorded as the lowest since 1774, and the year was also ranked top (in terms of

deficit rainfall) in the Great Alpine region (Haslinger and Blöschl (2017)), where the rainfall deficit began in winter 1920/21 and lasted until autumn 1921. In summary, the reconstructed runoff corresponded well to the majority of extreme drought years (e.g., 1540, 1616, 1669, 1710, 1921, as highlighted in Tab. 5) and previously demonstrated in the OWDA-based PDSI tree-ring reconstructions or other references (Wetter and Pfister, 2013; Cook et al., 2015; Dobrovolný et al., 2010; Brázdil et al., 2013; Markonis et al., 2018). This might be the case as the tree-ring proxies involved in the developed reconstruction were the same,

which could reveal the true nature of hydroclimatic shifts. Still, our reconstruction missed certain notable dry events, e.g., 1894 (Brodie, 1894) which was associated with unprecedented low levels of rainfall and excessive temperature rises in the south of England, the British Isles, and other European regions (Cook et al., 2015; Hanel et al., 2018; Brodie, 1894).





**Table 5.** Simulated runoff deficiency of extreme cases over past 500 years since 2000 CE

| station Name | No of events | simulated low flow years | minimum deficit (year) | maximum deficit(year) |
|---|---|---|---|---|
| Orsova-Danube | 12 | 1536, **1540**, **1669**, **1686**, 1704, 1706, **1710**, 1746, 1834, 1943, 1947, 1990 | 2.19 (1704) | 30.33(1686) |
| Dresden- Elbe | 1 | **1669** | | 2.76 (1669) |
| Wasserburg-Inn | 3 | **1669**, **1686**, 1754 | 1.79 (1754) | 27.8 (1669) |
| Blois-Loire | 17 | **1540**, 1603, 1631, 1634, **1669**, 1676, **1686**, 1706, **1710**, **1724**,1736, 1754, 1766, 1884, **1921**, 1945, 1949 | 0.07 (1766) | 85.7 (1669) |
| Montjean-Loire | 48 | **1540**, 1603, 1607, 1616, 1630, 1631, 1632, 1633, 1634, 1635, 1661, **1669**, 1670, 1676, 1680, 1681, 1684, 1685, **1686**, 1702, 1704, 1705, 1706, **1710**, 1715, 1717, 1718, 1723, **1724**, 1731, 1736, 1742, 1743, 1744, 1745, 1746, 1753, 1754, 1757, 1785, 1815, 1826, 1834, 1874, 1884, **1921**, 1945, 1949 | 1.95 (1874) | 105.2 (1686) |
| NeurHafen-Main | 6 | **1540**, **1669**, 1681, **1686**, 1706, **1724** | 0.26 (1681) | 83.84 (1669) |
| Wuerzburg-Main | 2 | **1540**, **1669** | 0.3 (1540) | 17.0 (1669) |
| BaselRheinhalle-Rhine | 21 | 1536, **1540**, 1590, 1603, 1616, 1631, 1666, **1669**, 1676, 1681, **1686**, 1704, 1706, **1710**, **1724**, 1736, 1746, 1753, 1754, **1921**, 1949 | 1.78 (1704) | 133.9 (1669) |
| Baselschifflaende-Rhine | 22 | 1536, **1540**, 1603, 1666, **1669**, 1676, **1686**, 1706, **1710**, **1724**, 1728, 1736, 1746, 1754, 1766, 1822, 1834, 1865, **1921**, 1947, 1949, 1976 | 3.60 (1766) | 370.8 (1669) |
| Köln-Rhine | 28 | 1536, **1540**, 1590, 1603, 1616, 1631, 1634, **1669**, 1676, 1681, 1684, **1686**, 1704, 1706, **1710**, **1724**, 1736, 1744, 1745, 1746, 1753, 1754, 1858, 1865, 1874, **1921**, 1949, 1976 | 1.34(1745) | 157.6 (1686) |
| Rees-Rhine | 18 | 1536, **1540**, 1603, 1631, 1666, **1669**, 1676, 1681, **1686**, 1704, 1706, **1710**, **1724**, 1736, 1746, 1754, **1921**, 1949 | 11.7 (1704) | 96.0 (1669) |
| Hann-Munden-Wesser | 11 | **1540**, **1669**, 1681, **1686**, 1706, **1710**, **1724**, 1911, 1934, 1976, 1991 | 1.95 (1991) | 46.6 (1669) |
| Bodenwerder-Wesser | 15 | **1540**, 1616, 1631, **1669**, 1681, **1686**, 1706, **1710**, **1724**, 1754, 1858, 1874, 1911, 1934, 1976 | 0.029 (1858) | 56.3 (1669) |
| Instchede- Wesser | 18 | **1540**, 1616, 1631, **1669**, 1670, 1676, 1681, 1685, **1686**, 1706, **1710**, 1754, 1814, 1857, 1858, 1865, 1934, 1959 | 0.30 (1670) | 134.4 (1669) |





## 5    Conclusions

In this study, hydrological (GR1A) and data-driven (BRNN, LSTM) models were used to simulate runoff during the period
1500–2000, considering various input fields. Different input configurations were evaluated for runoff predictions. Following
validation of the simulated series, we provided runoff reconstructions for 14 catchments across Europe (Germany: Main, Rhine,
Wesser, Inn, the Netherlands: Rhine and Romania: Danube). The main findings can be summarized as follows:

1. Data-driven methods have proven to be helpful for runoff simulations even when there are deficiencies in the driving input
   fields. This contrasts with a conceptual hydrological model, which would require bias correction before the simulation.

2. There is no significant difference between the BRNN and LSTM-simulated runoff neither in terms of the individual
   values nor in relation to the validation metrics.

3. Validation skill metrics suggest that for runoff prediction, it is beneficial to consider data-driven models that explicitly
   account for serial dependence either through input data (e.g., time-lagged input fields) or directly in the model structure
   (e.g., LSTM - networks).

4. The droughts identified in the reconstructed series correlated well with significant documented events (such as 1540,
   1669, and 1921).

The reconstructed series relies heavily on the consistency of underlying reconstructed precipitation (Pauling et al., 2006)
and temperature (Luterbacher et al., 2004) forcing fields. Unfortunately, this cannot be fully verified directly, due to the lack of
sufficient long-term observational data sets. With the limited information (GHCN), we identified several notable deficiencies
in the reconstructed forcings, in particular, underestimated variance in precipitation reconstruction, leading to inconsistencies
in observed runoff (e.g., demonstrated by the poor results of GR1A for some catchments). Moreover, proxy records are spa-
tially heterogeneous (also used in the development of gridded reconstructions). Due to the fact that some regions are better
represented than others and inevitably this results to poor performance over the latter.

However, the skill of precipitation and temperature reconstructions across the selected catchments to develop runoff is fairly
good. In addition, the data-driven methods that were used in the paper are capable of removing systematic bias (as was proven
in validation). We cannot be sure that the link between reconstructed forcing and runoff is stationary when going back in time.
Moreover, when the number of natural proxies decreases, the uncertainty increases. The reconstructed data should, therefore,
always be considered with caution. In addition, we showed that the skill of the reconstructed forcings decreases with time-scale.
This may imply problems with the representation of multi-year droughts.

Future research could consider further improvements of the simulations, e.g., by training a meta-model combining the runoff
simulations from several fitted models. Since interest is not often focused on the runoff series, but on some other indicator (such
as PDSI in the case of drought), it is also possible to simulate the drought indices directly, considering either the precipitation
and temperature input fields or the simulated runoff. Finally, discrete classifiers could also be used to simulate the drought (or
water level) classes directly.





## 6 Data Availability

The runoff reconstruction were prepared using the below data set and can be accessed at free, public repository Figshare (https://doi.org/10.6084/m9.figshare.15178107, Sadaf et al. 2021). The gridded data of precipitation and temperature can be downloaded at website via link https://www.ncdc.noaa.gov/data-access/paleoclimatology-data. The monthly global historical climatological network (GHCN) provides revision and updated version (V4) for temperature and (V2) precipitation which can be accessed via the link https://www1.ncdc.noaa.gov/pub/data/ghcn/. The data repositories of GRDC runoff is accessible for public at https://www.bafg.de/GRDC/EN/Home/homepage_node.html. .

## Appendix A

### A1 Goodness-of-fit assessment

We used a few statistical measures to assess the skillfulness of runoff reconstruction using a gridded-based simulation and an observed data-set. These measurements are mathematically defined as follows:

$$rSD = \frac{SD_{g_i}}{SD_{o_i}}$$

The terms $g_i$ and $o_i$ refer to the gridded and observed time series at point $i$, respectively. The standard deviations ratio (rSD) returns the maximum value of 1. The observed variability is underestimated when the value is less than one, while the observed variability is overestimated when the value is greater than one.

$$RMSE = \sqrt{\sum_{i=1}^{n} \frac{(g_i - o_i)^2}{n}}$$

$$MAE = \sum_{i=1}^{n} \frac{|(g_i - o_i)|}{n}$$

The RMSE and MAE measure how well predictions fit the measurements. MAE and RMSE values can range from 0 to infinity, indicating a perfect fit to a zero fit.

$$R = \frac{cor_{g_i}}{cor_{o_i}}$$

$Cor$ computed the correlation of observed and predicted data. The method can be specified as "kendall" or "spearman". Kendall's tau or Spearman's rho are used to estimate rank-based competence. The Nash–Sutcliffe efficiency (NSE), alternatively referred to as model efficiency (Nash and Sutcliffe, 1970), is a metric for the model's overall competence. It is defined as follows:

$$NSE = 1 - \frac{\sum_i (g_i - o_i)^2}{\sum (o_i - mean(o_i))^2}$$





NSE = 1 corresponds to a perfect match between predicted and observed data, while a value less than 0 indicates that model predictions are on average less accurate than using the long-term mean of the observed time series $mean(o_i)$.

Another coefficient of efficiency D, the index of agreement represents a decided improvement over the coefficient of determination but also is sensitive to extreme values, owing to the squared differences.

$$d = 1 - \frac{\sum_i (g_i - o_i)^2}{\sum(|g_i - mean(o)||o_i - mean(o_i)|)^2}$$

The index of agreement ranges from 0.0 to 1.0, with higher values signifying a better agreement between the model and observations, similar to the interpretation of the coefficient of determination.

$$KGE = 1 - ED \tag{A1}$$

$$ED = \sqrt{s[1] * (r-1)^2 + s[2] * (\alpha - 2)^2 + s[3] * (\beta - 2)^2} \tag{A2}$$

$$\alpha = \frac{\sigma_g}{\sigma_o} \tag{A3}$$


$$\beta = \frac{\mu_g}{\mu_o} \tag{A4}$$

The Kling-Gupta efficiency (KGE) index is calculated using three primary components: r, $\alpha$, and $\beta$. The symbol r denotes the Pearson product-moment correlation coefficient; $\alpha$ denotes the ratio of the standard deviations of the simulated and observed values; and $\beta$ denotes the ratio of the mean of the simulated and observed values. $\alpha, \beta$, and r have an ideal value of one. $s$ is a three-dimensional numeric representation of the scaling factors of length three that is used to adjust the relative importance of various components.


### A2    Data prepossessing of LSTM

To build the LSTM model, we use the Keras environment with its high-level application programming interface (API) for neural networks and tensor flows. Figure A1 represents the structure of the LSTM neural model for the rainfall runoff relationship in several catchments. We design our network by stacking one LSTM and two dense layers on top of one other. As shown in Fig.

A1, the model configured four distinct input combinations, each of which was normalized to [0, 1] in the training and testing phases. The model considers the Rectified Linear Unit (ReLU), using component wise multiplication and defining the dropout parameter as 0.1. According to Kingma and Ba (2014), the optimization algorithm plays a significant role in the algorithm's convergence and optimization. For this reason, Adam's optimizer is considered, as it performs stochastic gradient descent (SGD) more efficiently using the backpropagation algorithm. During compilation, the learning rate is set to '0.001' or '0.002'

and the model selects random batch sizes and epochs. In addition, the mean absolute error is a function used as an objective to minimize residues and achieve optimum value. The checkpoint algorithm is also applied to test the model's accuracy level. Finally, the best output of the model is saved, with minimum loss and better accuracy.





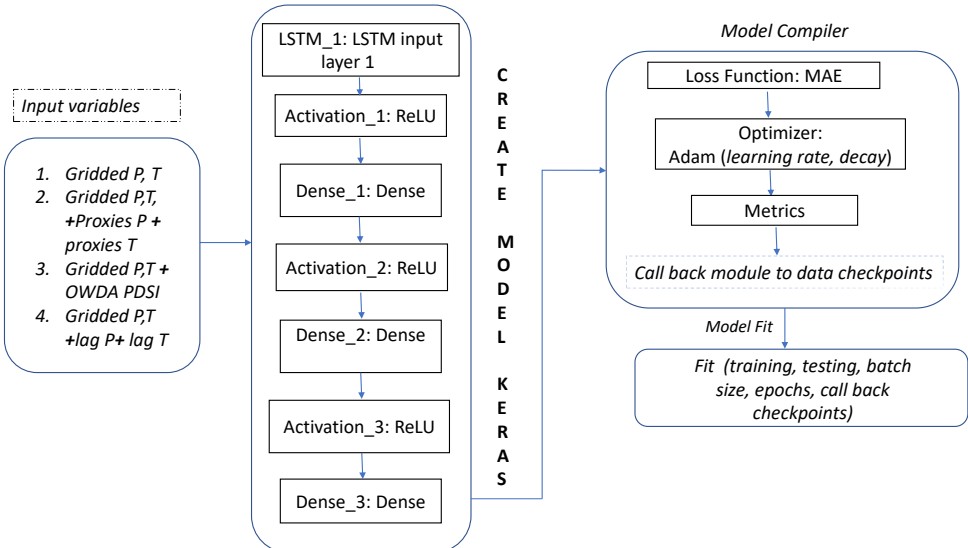

**Figure A1.** Structure of LSTM neural network model in KERAS environment for runoff predictions

*Author contributions.* The study was initially designed by RK, MH and YM. Algorithms are coded with the assistance of YM, US and MH. Datasets were collected by VG and SN. The research was carried out by SN, MS, and MH, who also wrote the paper. OR and RK both helped
to revise the manuscript.

*Competing interests.* The authors declare that they have no conflict of interest.

*Acknowledgements.* This work was carried out within the bilateral project XEROS (eXtreme EuRopean drOughtS: multimodel synthesis of past, present and future events), funded by Czech Science Foundation (Grant No. 1924089J) and the Deutsche Forschungsgemeinschaft (Grant No. RA 3235/11) + IGA (Project No.2020B0018). We thank the Global Runoff Data Centre (GRDC) for providing the observed
runoff data. All analyses and visualisations were done using R.





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
