# Peer review of "A 500-year annual runoff reconstruction for 14 selected European catchments"

_Earth System Science Data, 2021_

## Referee Comment (RC1)

The manuscript by Nasreen et al., investigate the reconstruction of annual runoff timeseries for 14 European catchments over the period 1500-2000. In the first part, the authors evaluate the validity of an existing precipitation and temperature reconstruction dataset against GCHN stations. In a second step, they evaluate the use of 2 data-driven models and a lumped hydrological models to predict annual runoff. In a third section, they provide an overview on years with low annual runoff occurred in the selected 14 European catchments during the last 500 years.

Main comments:

I believe the manuscript need improvements in the methodology description, some additional analysis especially related to model and input forcing evaluation, as well as a bit of text reorganization and polishing in some sections. Here below I list my comments.

Important remarks

1. The authors should make it clear through the entire text that the manuscript deal with annual runoff reconstructions. Neither the title and the abstract mention it. I would suggest starting by modifying the title with "A 500-year annual runoff reconstruction for 14 selected European catchments".
2. Instead of "long-term", I would suggest using the term "multi-century"
3. When speaking about droughts in the text, please always specify the scale of the considered drought:
   - Runoff drought → Annual runoff droughts
   - Drought duration → Multi-year drought duration
   - Runoff drought severity → Annual runoff drought severity
4. I wonder if it's wort to keep in the text everything related to the "natural proxy data".
   - The inclusion of such data does not improve the model at all. You could simply add a sentence in the model discussion saying that "the inclusion of additionally proxy data has been investigated but did not provide benefits to model accuracy". I would suggest to just focus on the improvement provided by adding drought indicator (scPDSI).
   - In the text you mention multiple times the use of "natural proxy data" that clutter and complicate the reading in the introduction, Section 2.2 and 3.1.
   - In Section 2.2. you speak about data standardization. For what reason? Then you applied the normalization to 0-1 for model training as described in the Appendix?
   - For GRDC stations where there is no close proxy data which data did you use? There is skill reported for all catchments in Table 3 for "Gridded+Proxy" models
   - Are you adding a column for the precipitation natural proxy, and one column for the temperature proxy? This is not explained in the text.
   - In Section 3.1 you say: "selected the raw proxy data from inside the catchment or within a 100 km buffer around the catchment.". The following question arises:
     ➔ If more than one proxy in the 100 km outside the catchment do you take the average value of the proxies?
     ➔ If no proxy in the 100 km radius, which value do you assign to the catchment?
     ➔ Is it representative a single proxy within a catchments that extends thousands of $km^2$ (tens of 0.5 x 0.5 grid cells)?

5. I would suggest creating a separate section to introduce the scPDSI drought indicator (new Section 2.2?).
6. In Section 3.1 please introduce how you define the calibration and validation set. Currently they are defined only in the results Section 4.2
7. Clearly state that GR1A is a conceptual lumped hydrological model.
8. In Section 3.2, please specify in more detail how the X parameter of GR1A is optimized and if it is optimized independently for each catchment. Only in the result Section 4.2 I can read "for each catchment separately".
9. In Section 3.3, please specify if a NN model is trained for each catchment, or a single model is trained for all catchments
10. In Section 3.3, I would avoid the use of term "Gridded". All models received as input is the sum/average catchment P and T. Maybe the word "Forcing" is more appropriate. "Gridded" erroneously make thinking to "distributed" or gridded simulations.
    Additionally, at line 159, please specify that the "lagged forcing" refers to 1-year lag data. Currently is just specified at line 259. Also provide explanation why you didn't use additional temporal lags (i.e. 2 and 3 years).
11. In Section 3.3 please clarify what is currently described at line 175-179.
    - "Best performance" at L175 refers to which metric? MAE?
    - "To reduce the likelihood of overfitting during the calibration/training, a fraction of the calibration data was used to check the performance of an independent (or so-called "testing") set"
      ➔ Which fraction?
      ➔ I am confused. Training/Calibration: 1900-2000; Verification/Test set: (prior 1900). The model tuning/validation set is a fraction of 1900-200 data?
      ➔ Please use the term "test set" only for data not used for model training and hyperparameter tuning
12. L12: "On the other hand, the data-driven models have been proven to correct this bias (referred to underestimation of variance)".
    In the main text, but also in the supplementary you don't provide the rSD metric on the annual runoff evaluation against GRDC. It is therefore difficult to verify such statement. On my experience, data-driven models are good in coping with conditional bias in the data, less in conditional variance. I would be surprised if you overestimate the variability of runoff. Please provide the rSD metric also in the runoff evaluation. In Fig.5, I see all models to underestimate the variance!
13. What you define as rSD in the Appendix is the "ratio of standard deviations" and not the "relative error in standard deviation" as referred to at line 181.
14. In both the evaluation of P, T and R, you don't provide information related the bias. Please provide the BIAS (mean difference between pred. and obs.) or the relative BIAS (BIAS/mean(obs)). You could report BIAS instead of D.
15. Please correct the definition of the skill metrics in the appendix.
    - The definition of R at line 404 is wrong!
    - At line 412, the coefficient of determination is equivalent to $R^2$. And "decided improvement" is maybe a too strong word …

- In the equation of the index of agreement (maybe IoD), which sometimes appears as D and sometimes as d, in the denominator there is a missing "i" subscript within mean(o).
- At line 421, alpha corresponds to rSD, r to R. There is lot of repetitions. I guess you could remove also the scaling factors "s" within KGE since I guess you use 1 for all of them.
- Maybe add the BIAS or relBIAS metric

16. I would suggest removing entirely the analysis of the impact of aggregating ed time-scale analysis. I believe it does not have anything in relation to the objective of the manuscript, and introduce plenty of questionable sentence
    - L220: "The RMSE decrease with increasing temporal aggregation because the RMSE depend on the number of observations. "
    → I would eventually argue that RMSE decrease because aggregating over time smooth (aka) decrease the variance.
    - L222: "Except for correlation which shows relatively stable values over aggregations, it is evident that the reconstruction skill decreases the greater the (aggregation) time-scale".
    → It means that for the reconstruction skill you refer to NSE or KGE.
    → The rSD is expected to decrease when averaging over time
    → If NSE and KGE decreases, if the correlation is relatively stable, and the RMSE decrease, the source of the decrease is increasing bias. But you don't provide information on it …
    - Eventually, the caption of Figure 4 should be completely reformulated. It does not describe the figure content, it does not mention if it refers to P or T evaluation; it refers to GRDC instead of GHCN, ..
    → "Fig. 4. Benchmarking GOF accuracy of P (or T) reconstruction against GHCN stations at various temporal scale …
17. Figure 2 and 3 should be revised.
    - Please add the BIAS or relBIAS metric (eventually replacing IoD)
    - Please correct the colorbar limits to facilitate comparison. I suggest setting to 1 the max value for Index of Agreement, NSE and KGE colormaps.
    - KGE should be bounded to 0 as far as I know. But I see negative values !!!
    - NSE below 0 means that the long-term mean of the station time series would provide better accuracy than using the reconstruction. Maybe set lower limits of NSE also to 0 (unbounded left)
    - Strangely the KGE colormap as a single step value of 0.3 (0.3-0.6). I guess there is a code mistake here !!!
    - Slightly reduce the marker size to reduce a bit the superimpositions of the circles.
18. Please color code the cells of Table 3 with the same colormaps of Fig 2 and 3
19. I am not sure I understand what is represented in Fig. 8. Is a comparison between GRDC vs simulated runoff values in the Q0-Q33 range? If yes, the axis label should be runoff [mm/year] !!!
20. I find really interesting the analysis in 4.4. Maybe you could highlight the value of some your statements, by adding an interesting figure or by for example plotting some drought years labels close to their cdf points in Fig 6.

21. I think that an additional plot with the "best" reconstruction of one or two time series (selected and zoomed) from Fig S1 and S2) could be a nice addition to the main manuscript.

22. I would like to draw your attention to the fact that there are a couple of works related to century and multi-century hydrological reconstructions that are not currently present in your references and would be worth adding:

   - Caillouet, L., Vidal, J.-P., Sauquet, E., Devers, A., and Graff, B.: Ensemble reconstruction of spatio-temporal extreme low-flow events in France since 1871, Hydrol. Earth Syst. Sci., 21, 2923–2951, https://doi.org/10.5194/hess-21-2923-2017, 2017.
   - Ghiggi, G., Humphrey, V., Seneviratne, S. I., and Gudmundsson, L.: GRUN: an observation-based global gridded runoff dataset from 1902 to 2014, Earth Syst. Sci. Data, 11, 1655–1674, https://doi.org/10.5194/essd-11-1655-2019, 2019.
   - Ghiggi, G., Humphrey, V., Seneviratne, S. I., & Gudmundsson, L. (2021). G-RUN ENSEMBLE: A multi-forcing observation-based global runoff reanalysis. Water Resources Research, 57, e2020WR028787. https://doi.org/10.1029/2020WR028787
   - Moravec, V., Markonis, Y., Rakovec, O., Kumar, R., & Hanel, M.(2019). A 250-year European drought inventory derived from ensemble hydrologic modeling. Geophysical Research Letters, 46. https://doi.org/10. 1029/2019GL082783
   - Smith, K. A., Barker, L. J., Tanguy, M., Parry, S., Harrigan, S., Legg, T. P., Prudhomme, C., and Hannaford, J.: A multi-objective ensemble approach to hydrological modelling in the UK: an application to historic drought reconstruction, Hydrol. Earth Syst. Sci., 23, 3247–3268, https://doi.org/10.5194/hess-23-3247-2019, 2019.

   Related to Rhine Drought, this one is also very interesting:

   Christian Pfister, Rolf Weingartner & Jürg Luterbacher (2006) Hydrological winter droughts over the last 450 years in the Upper Rhine basin: a methodological approach, Hydrological Sciences Journal, 51:5, 966-985, DOI: 10.1623/hysj.51.5.966

23. Facultative (but potentially interesting and very appreciated), I would be curious to see how a temporally aggregated century-long monthly runoff reconstruction such G-RUN (Ghiggi et al., 2019, 2021) (i.e. forced by GSWP3) would compare to your annual time series during the calibration period. I guess it could require a couple of day of work, but I am intrigued to know if an ad-hoc catchment based annual runoff reconstruction provides better results than annual catchment runoff derived from gridded monthly runoff timeseries.

Minor corrections
7: long-term → multi-century annual runoff reconstructions are still lacking (…)
7: Remove: (e.g. monthly, …. )
9: Remove: proxy data (if you follow Important Remark 3)
25: For the last 40 years → In the last 40 years
26: Missing reference for the 45 million loss fact
30-33: To be reformulated, please!

43-44: To be reformulated, please!

47-48: To be reformulated, please!

51: Reference Breiman et al., 2001 do not refer to a ML application for runoff/streamflow forecasting. You can find better ones ☺

51: Reference Thiesen et al., 2019 and "shannon entropy" are not used for runoff/streamflow forecasting

53: Contrasting (changing) → Changing

53: Suggestion: limit their application outside boundary conditions observed during model training.

55: long-term → multi-century annual runoff reconstruction for 14 European catchments

57: Remove: proxy data (if you follow Important Remark 3)

57: "other long-term historical data sources" → GRDC and scPDSI

57: we use a combination of → we benchmarked the use of

58: Conceptual HM → Conceptual lumped HM

59: annual evolution → annual runoff

59: "We pay particular attention to low flows during drought years."
    The models are not optimized to pay particular attention to negative annual runoff anomalies so I would avoid such sentence.

60: "Using long-term data on climatic conditions and runoff may provide an efficient technique of visualizing droughts and low flow periods". Please reformulate or remove.

63: Drought identification → Drought identification methodology

63. To be reformulated. Suggestion: The accuracy of the employed precipitation and temperature reconstructions, as well as the derived runoff simulations, is evaluated in Section

69: data from → scPDSI drought indicator data from

69: Remove: natural proxies (if you follow Important Remark 3)

75: To be reformulated, please!

76. What about subparagraph: 2.1.1 Precipitation, 2.1.2 Temperature?

94-96: Consistency: Choose between dataset or data-set

104: "The runoff series from the GRDC were selected based on the condition of data availability, at least 25 years prior to 1900." → Only GRDC runoff time series with at least 25 years of data prior 1900 were selected

124: Remove "we" → Section 3.4 …

125: Section 3.5 presents the methods to identify annual runoff droughts

132: and the proxy data and

133: validation of individual catchments (Fig.2) → Fig 2 refers to P evaluation ☹

135: See Important remark 4

181: Provide the metrics in Capital Case format

199-204: This maybe belong more to Section 3.2 and 3.3

212: "Some stations indicated a worse performance and could not adequately capture the observed temperature variability". → Very likely, is not the station that has bad skill, but the reconstruction☺ → "Low skill observed at some GHCN stations can be explained by the unresolved variability of grid-cell average temperature, especially in regions with complex-terrain."

217: Consistency: GOF or gof

236-239: Move to Section 3.1

240-241: GR1A is not driven by gridded data, but the catchment average value … ! Maybe move to Section 3.2.

260: Please reformulate (or remove between brackets content)

277: I don't get how scPDSI provide better representation of the temporal dependency structure

287-288: Please reformulate

294: Please reformulate

297:  simulations → cumulative distribution of simulated runoff values

319: match, less → agreement, lower

354, 358, 362,374,376: runoff → annual runoff

359: conceptual → conceptual lumped

371-373: Maybe remove?

374: develop → derive

378-379: I suggest removing the aggregation time scale analysis and results

396. Specify g and o before starting describing the metrics.

395: measurement → metrics

396: ratio of standard deviations

435: Remove: and epochs.

435-437: Suggestion: The Huber Loss is employed to minimize the mean absolute error between observations and predictions. Model checkpointing is used to keep track of model weights evolutions during training and select the best model weights when the allocated max number of training iterations is reached.

For further questions / discussions:
Gionata Ghiggi: gionata.ghiggi@epfl.ch

---

## Author Comment (AC1)

**Reply to Reviewers**

Dear Dr. Ghiggi, thank you for your constructive and valuable comments. We have revised the manuscript in response to your comments and hope that you will find our revised manuscript suitable for publication.

**Reviewer 1**

The manuscript by Nasreen et al., investigate the reconstruction of annual runoff timeseries for 14 European catchments over the period 1500-2000. In the first part, the authors evaluate the validity of an existing precipitation and temperature reconstruction dataset against GCHN stations. In a second step, they evaluate the use of 2 data-driven models and a lumped hydrological model to predict annual runoff. In a third section, they provide an overview on years with low annual runoff occurred in the selected 14 European catchments during the last 500 years.

**Major Comments**

1. The authors should make it clear through the entire text that the manuscript deal with annual runoff reconstructions. Neither the title and the abstract mention it. I would suggest starting by modifying the title with "A 500-year annual runoff reconstruction for 14 select-ed European catchments".

   **Author's Response:** We agreed and have updated the title, abstract **(L10)** as well as other occurrences throughout the text.
   **Title**: "A 500-year annual runoff reconstruction for 14 selected European catchments".
   The abstract was modified as,
   **L7-10 ➔** "In this study, we have used reconstructed precipitation and temperature data, Palmer Drought Severity Index and available observed runoff across fourteen European catchments in order to develop annual runoff reconstructions for the period 1500–2000 using two data-driven and one conceptual lumped hydrological model."

2. Instead of "long-term", I would suggest using the term "multi-century"

   **Author's Response: L7**, long-term was changed to multi-century.

3. When speaking about droughts in the text, please always specify the scale of the considered drought:

   - Runoff drought ➔ Annual runoff droughts
   - Drought duration ➔ Multi-year drought duration
   - Runoff drought severity ➔ Annual runoff drought severity

   **Author Response:** We agreed and hence, we have made changes throughout the manuscript (e.g. **L190 and L195).**

4. I wonder if it's wort to keep in the text everything related to the "natural proxy data".

   - The inclusion of such data does not improve the model at all. You could simply add a sentence in the model discussion saying that "the inclusion of additionally proxy data has been investigated but did not provide benefits to model accuracy". I would suggest to just focus on the improvement provided by adding drought indicator (scPDSI).

   - In the text you mention multiple times the use of "natural proxy data" that clutter and complicate the reading in the introduction, Section 2.2 and 3.1.

   - In Section 2.2. you speak about data standardization. For what reason? Then you applied the normalization to 0-1 for model training as described in the Appendix?

- For GRDC stations where there is no close proxy data which data did you use? There is skill reported for all catchments in Table 3 for "Gridded+Proxy" models

- Are you adding a column for the precipitation natural proxy, and one column for the temperature proxy? This is not explained in the text.

- In Section 3.1 you say: "selected the raw proxy data from inside the catchment or within a 100 km buffer around the catchment.". The following question arises:

➔ If more than one proxy in the 100 km outside the catchment do you take the average value of the proxies?

➔ If no proxy in the 100 km radius, which value do you assign to the catchment?

➔ Is it representative a single proxy within a catchment that extends thousands of km2 (tens of 0.5 x 0.5 grid cells)?

**Author's Response:** Thank you for this suggestion. We agreed that it is much easier to present the analysis without other natural proxies for the reasons you are summarizing. Therefore, we removed all proxy-related material and analysis from the manuscript.

5. I would suggest creating a separate section to introduce the scPDSI drought indicator (new Section 2.2?).

   **Author's Response:** Following the comment, we introduced a new section.
   **L82 ➔** Section 2.2 'scPDSI Drought indicator'

6. In Section 3.1 please introduce how you define the calibration and validation set. Currently they are defined only in the results Section 4.2

   **Author's Response:** Indeed, thank you for this comment. We defined the calibration and validation periods in the beginning of the Methods section.
   **L129-130 ➔** "Data were split into two parts: calibration (1900–2000) and validation (<=1900) to assess the model's accuracy and to select an appropriate model."

7. Clearly state that GR1A is a conceptual lumped hydrological model.

   **Author's Response:** This was done:
   **L134-135 ➔** "The GR1A is a conceptual lumped hydrologic model Manabe (1969), considering dynamic storage and antecedent precipitation conditions."

8. In Section 3.2, please specify in more detail how the X parameter of GR1A is optimized and if it is optimized independently for each catchment. Only in the result Section 4.2 I can read "for each catchment separately".

   **Author's Response**: We updated the section 3.2.
   **L138-139 ➔** "The parameter $X$ is optimized individually for each catchment by maximizing the Nash-Sutcliffe efficiency (NSE) between observed and simulated runoff."

9. In Section 3.3, please specify if a NN model is trained for each catchment, or a single model is trained for all catchments

   **Author's Response:** The text was altered in response to the reviewer suggestion:
   **L174-176 ➔** "The model development process was repeated several times, minimizing the Root Mean Square Error (BRNN) and Mean Square Error (LSTM) for each catchment individually."

10. In Section 3.3, I would avoid the use of term "Gridded". All models received as input is the sum/average catchment P and T. Maybe the word "Forcing" is more appropriate. "Gridded" erroneously make thinking to "distributed" or gridded simulations.
    Additionally, at line 159, please specify that the "lagged forcing" refers to 1-year lag data. Currently is just specified at line 259. Also provide explanation why you didn't use additional temporal lags (i.e. 2 and 3 years).

**Author's Response:** We appreciated the comment because it pointed out some inaccuracies in our methodology description. Indeed, we did not use the actual gridded simulation but the mean value of P and T across the catchment (as now explicitly mentioned in the methods). We have decided to use lag 1 year for our analysis because the correlation between (lagged) annual precipitation and runoff drops to 0 after lag 1.

**L152-157 →** "We considered combinations of reconstructed forcing, OWDA-based scPDSI, and lagged forcing as an input into the network for both model types. Specifically, the network using only reconstructed forcing is referred to as "P+T", the network with reconstructed forcing and OWDA scPDSI is termed as "P+T+PDSI" and finally the network which includes 1-year lagged forcing is referred to as "P+T+Lag". Please note, that dependence between annual precipitation and runoff at longer time lags was explored as well but since the correlation drops significantly at lags longer than 1, longer lags were not considered in the models."

11. In Section 3.3 please clarify what is currently described at line 175-179.
    - "Best performance" at L175 refers to which metric? MAE?
    - "To reduce the likelihood of overfitting during the calibration/training, a fraction of the calibration data was used to check the performance of an independent (or so-called "testing") set"
    → Which fraction?
    → I am confused. Training/Calibration: 1900-2000; Verification/Test set: (prior 1900). The model tuning/validation set is a fraction of 1900-200 data?
    → Please use the term "test set" only for data not used for model training and hyperparameter tuning

    **Author's Response:** In LSTM method, Mean Absolute Error (MAE) was referred as the loss function and Mean Square Error (MSE) was used to check the internal measure of accuracy. In addition, the training data set ranged from 1900 to 2000 and the validation set spanned the years prior to 1800 until GRDC-Runoff became accessible. While, testing set (25% of trained data) was used to avoid overfitting and determine, when training became halted.
    The text was modified accordingly,
    **L172-176 →** "To set the optimal hyperparameters of the models (such as the number of neurons and activation functions, etc.) and to reduce the likelihood of overfitting during the calibration/training, the performance was checked considering an independent (or so-called "testing") set. This was pulled from the calibration data (1900–2000) as a (random) fraction (25%). The model development process was repeated several times, minimizing the Root mean square Error (BRNN) and Mean Square Error (LSTM) for each catchment individually. The model with the best performance was then chosen for further evaluation. "

12. L12: "On the other hand, the data-driven models have been proven to correct this bias (referred to underestimation of variance)". In the main text, but also in the supplementary you don't provide the rSD metric on the annual runoff evaluation against GRDC. It is therefore difficult to verify such statement. On my experience, data-driven models are good in coping with conditional bias in the data, less in conditional variance. I would be surprised if you overestimate the variability of runoff. Please provide the rSD metric also in the runoff evaluation. In Fig.5, I see all models to underestimate the variance!

    **Author's Response:** The rSD was added into the heatmap and runoff reconstructions in the Supplementary material. You are right - our statement in the previous version was incorrect - the variance is underestimated as clear now from the figure. We revised the text accordingly.
    **L11-13 →** "On the other hand, the validation of input precipitation fields revealed an underestimation of the variance across most of Europe, which is propagated into the reconstructed runoff series."

13. What you define as rSD in the Appendix is the "ratio of standard deviations" and not the "relative error in standard deviation" as referred to at line 181.

    **Author's Response: L179 →** The term "relative error in standard deviation" has been replaced by "Standard Deviation Ratio" in the Appendix.

14. In both the evaluation of P, T and R, you don't provide information related the bias. Please provide the BIAS (mean difference between pred. and obs.) or the relative BIAS (BIAS/mean(obs)). You could report BIAS instead of D.

**Author's Response:** We agreed. The D metric figures were replaced by BIAS or relBIAS figures.

15. Please correct the definition of the skill metrics in the appendix. - The definition of R at line 404 is wrong!
- At line 412, the coefficient of determination is equivalent to R2 . And "decided improvement" is maybe a too strong word . . .
- In the equation of the index of agreement (maybe IoD), which sometimes appears as D and sometimes as d, in the denominator there is a missing "i" subscript within mean(o).
– At line 421, alpha corresponds to rSD, r to R. There is lot of repetitions. I guess you could remove also the scaling factors "s" within KGE since I guess you use 1 for all of them.
- Maybe add the BIAS or relBIAS metric

**Author's Response:** The definition of R was corrected and added at lines ➜ L405-406.

$$R = \frac{\sum_{i=1}^{n}(p_i - \overline{p})(o_i - \overline{o})}{\sqrt{\sum_{i=1}^{n}(p_i - \overline{p})^2}\sqrt{\sum_{i=1}^{n}(o_i - \overline{o})^2}}$$

Where "p" and "o" referred to predicted and observed value. The definition of D was deleted, since it is not reported anymore. The KGE was simplified and scaling factors have been replaced with 1. In addition, the BIAS and relBIAS metrics were included in Appendix.

16. I would suggest removing entirely the analysis of the impact of aggregating ed time-scale analysis. I believe it does not have anything in relation to the objective of the manuscript, and introduce plenty of questionable sentence
- L220: "The RMSE decrease with increasing temporal aggregation because the RMSE depend on the number of observations. "
➜ I would eventually argue that RMSE decrease because aggregating over time smooth (aka) decrease the variance.
- L222: "Except for correlation which shows relatively stable values over aggregations, it is evident that the reconstruction skill decreases the greater the (aggregation) time-scale".
➜ It means that for the reconstruction skill you refer to NSE or KGE.
➜ The rSD is expected to decrease when averaging over time
➜ If NSE and KGE decreases, if the correlation is relatively stable, and the RMSE decrease, the source of the decrease is increasing bias. But you don't provide information on it . . .
- Eventually, the caption of Figure 4 should be completely reformulated. It does not describe the figure content, it does not mention if it refers to P or T evaluation; it refers to GRDC instead of GHCN, ..
➜ "Fig. 4. Benchmarking GOF accuracy of P (or T) reconstruction against GHCN stations at various temporal scale . . .

**Author's Response:** Thank you for this suggestion. This analysis originated in the preliminary exploration of the dataset and we agreed that it is not needed for the scope of the paper. The multi-scale analysis was therefore, removed from the paper.

17. Figure 2 and 3 should be revised. - Please add the BIAS or relBIAS metric (eventually replacing IoD)
- Please correct the colorbar limits to facilitate comparison. I suggest setting to 1 the max value for Index of Agreement, NSE and KGE colormaps.
- KGE should be bounded to 0 as far as I know. But I see negative values!!!
- NSE below 0 means that the long-term mean of the station time series would provide better accuracy than using the reconstruction. Maybe set lower limits of NSE also to 0 (unbounded left)
- Strangely the KGE colormap as a single step value of 0.3 (0.3-0.6). I guess there is a code mistake here!!!
- Slightly reduce the marker size to reduce a bit the superimpositions of the circles.

**Author's Response:** Figures 2 and 3 were renamed as Figures 3 and 4, respectively. Hereafter, Relative bias is included in Figure 3. While for Figure 4, we introduced Bias instead of D. In both cases, the "relative error of standard deviation"

was replaced with the "Standard deviation ratio". Furthermore, we reduced the marker size and adjusted the KGE and NSE thresholds from maximum to minimum ranges and corrected all color scales.

18. Please color code the cells of Table 3 with the same colormaps of Fig 2 and 3

**Author's Response:** The colormap of Table 3 was updated as similar to Figures 2 and 3. Also, Table 3 was altered to Figure 4, when the legend color bar was added. Likewise, the Supplementary tables were changed.

19. I am not sure I understand what is represented in Fig. 8. Is a comparison between GRDC vs simulated runoff values in the Q0-Q33 range? If yes, the axis label should be runoff [mm/year] !!!

**Author's Response:** Figure 8 originally represents the runoff deficit based on 33% threshold. As a result, the Figure labels were written as runoff deficit[mm/year], and the caption was also modified.
"The observed and simulated runoff deficit based on the 33rd percentile threshold for 14 selected catchments during the calibration and validation period."

20. I find really interesting the analysis in 4.4. Maybe you could highlight the value of some your statements, by adding an interesting figure or by for example plotting some drought years labels close to their cdf points in Fig 6

**Author's Response:** "We appreciate Reviewer remarks. Figure 6 was modified by adding drought events to each panel and further lines were added.
**L282-286 ➔** "The most severe drought year identified by the models was the same in the periods 1500–1800 and 1900–2000 (Figure 7 left and right panels), while for 1800–1900 the models identified either 1865 (GR1A, LSTM) or 1858 (BRNN, 2nd worse for LSTM). Please note that the 1858 low water mark is available at Laufenburg Pfister et al. (2006) near Basel and was regarded as one of the worst winter droughts in the last 200 years."

21. I think that an additional plot with the "best" reconstruction of one or two time series (selected and zoomed) from Fig S1 and S2) could be a nice addition to the main manuscript.

**Author's Response:** We agreed, and Figure 8 is included in the main text to show the two best reconstructed runoff series.

22. I would like to draw your attention to the fact that there are a couple of works related to century and multi-century hydrological reconstructions that are not currently present in your references and would be worth adding:
- Caillouet, L., Vidal, J.-P., Sauquet, E., Devers, A., and Graff, B.: Ensemble reconstruction of spatio-temporal extreme low-flow events in France since 1871, Hydrol. Earth Syst. Sci., 21, 2923–2951, https://doi.org/10.5194/hess-21-2923-2017, 2017.
- Ghiggi, G., Humphrey, V., Seneviratne, S. I., and Gudmundsson, L.: GRUN: an observation-based global gridded runoff dataset from 1902 to 2014, Earth Syst. Sci. Data, 11, 1655–1674, https://doi.org/10.5194/essd-11-1655-2019, 2019.
- Ghiggi, G., Humphrey, V., Seneviratne, S. I., Gudmundsson, L. (2021). G-RUN ENSEMBLE: A multi-forcing observation-based global runoff reanalysis. Water Resources Research, 57, e2020WR028787. https://doi.org/10.1029/2020WR028787
- Moravec, V., Markonis, Y.,Rakovec, O., Kumar, R., Hanel, M.(2019). A 250- year European drought inventory derived from ensemble hydrologic modeling. Geophysical Research Letters, 46. https://doi.org/10. 1029/2019GL082783
- Smith, K. A., Barker, L. J., Tanguy, M., Parry, S., Harrigan, S., Legg, T. P., Prudhomme, C., and Hannaford, J.: A multi-objective ensemble approach to hydrological modelling in the UK: an application to historic drought reconstruction, Hydrol. Earth Syst. Sci., 23, 3247–3268, https://doi.org/10.5194/hess-23-3247-2019, 2019.
Related to Rhine Drought, this one is also very interesting: Christian Pfister, Rolf Weingartner Jürg Luterbacher (2006) Hydrological winter droughts over the last 450 years in the Upper Rhine basin: a methodological approach, Hydrological Sciences Journal, 51:5, 966-985, DOI: 10.1623/hysj.51.5.966

**Author's Response:** The reference (Moravec et al., 2019) already exists in the manuscript. The remaining suggested references were added to the database and the following text was updated in the subsection.
**L42-44 ➔** "As another example, Caillouet et al. (2017) provides a 140-year data set of reconstructed streamflow over 662 natural catchments in France since 1871 using the GR6J hydrological model, highlighting several well-known extreme

low flow events."

**L341-343 ➜** "Monthly runoff anomalies analyzed from the GRUN data set (Ghiggi et al., 2019) show that August 1976 was the fifth driest month between 1900 and 2014, with some of our study catchment also signaling the 1976 yearly drought (e.g Köln-Rhine, Hann-Munden-Wesser, Bodenwerder-Wesser)."

**L44-46 ➜** "A multi ensemble modeling approach using GR4J has been applied by Smith et al. (2019) to develop a UK-based historical river flows and examine the potential of reconstruction for capturing peak and low flow events from 1891 to 2015."

**L284-286 ➜** "Please note that the 1858 low water mark is available at Laufenburg Pfister et al. (2006) near Basel and was regarded as one of the worst winter droughts in the last 200 years."

23. Facultative (but potentially interesting and very appreciated), I would be curious to see how a temporally aggregated century-long monthly runoff reconstruction such GRUN (Ghiggi et al., 2019, 2021) (i.e. forced by GSWP3) would compare to your annual time series during the calibration period. I guess it could require a couple of day of work, but I am intrigued to know if an ad-hoc catchment based annual runoff reconstruction provides better results than annual catchment runoff derived from gridded monthly runoff time series.

**Author's Response:** The statistical analysis of annual-based GRUN forced by GSWP3 and runoff reconstruction against GRDC runoff (Figure S9) and two time series (Figure S10) were included in the Supplementary Material and are also presented below.

[Figure]

**Figure 1.** Statistical comparison between reconstructed and GRUN runoff with respect to observed GRDC Runoff for the common period 1902-2000

[Figure]

**Figure 2.** The simulated, observed, and GRUN time series for Dresden and Weurzburg catchments from 1900 to 2000. Model comparisons in terms of statistics can be seen at the top of the Figure.

Text was also added in the manuscript,
**L293-300** ➜ "Finally, to check to consistency of our reconstructed dataset, we compared the skill of our simulation with respect to the GRDC runoff observation with that of the GSWP3-forced GRUN monthly runoff averaged over the catchments (Supplementary Fig. S9 and S10). Our reconstruction outperforms GRUN data in RMSE, MAE, relBIAS and NSE in the majority of the catchments, while the correlation to GRDC runoff is slightly higher for GRUN compared to our reconstruction. The variability, which is underestimated by our data-driven models (on average by 16.5%) is over-estimated by GRUN (on average by 17.2%). Since the correlation compensates the bias the KGE for our reconstruction and GRUN is comparable. This suggests that GRUN could be used for data-driven model training, provided at least some information on flow characteristics is available in the catchment."

**Minor Comments**

7: long-term ⟶ multi-century annual runoff reconstructions are still lacking (. . . )

7: Remove: (e.g. monthly, . . . . )

9: Remove: proxy data (if you follow Important Remark 3)

Author's Response: L8 ➜ The text was removed

25: For the last 40 years ⟶ In the last 40 years

Author's Response: L23-24 ➜ The text was updated accordingly

26: Missing reference for the 45 million loss fact

Author's Response: L24 ➜ Reference was added (Anonymous, 2020).

30-33: To be reformulated, please!

**Author's Response:** "Continuous records of runoff/discharge series are no longer available, including various multi-year droughts and pluvial periods. On the other hand, proxy-based (typically seasonal or annual) reconstructions are alternatively used, considering various proxy data, such as past tree-rings (Cook et al., 2015; Casas-Gómez et al., 2020; Tejedor et al., 2016; Kress et al., 2010; Nicault et al., 2008), speleothem (Vansteenberge et al., 2016), ice cores, sediments (Luoto and Nevalainen, 2017)"
**L25-32** has been changed to,
"While runoff is a key element related to water security, it is difficult to interpret recent hydroclimate fluctuations (multi-year droughts in particular) considering observed runoff records (Markonis and Koutsoyiannis, 2016; Hanel et al., 2018), which are in general seldom available for years prior 1900. In this way, we are missing runoff information on various severe multi-year droughts and pluvial periods, which can be assessed only indirectly using (typically seasonal or annual) reconstructions based on various proxy data, such as past tree-rings (Nicault et al., 2008; Kress et al., 2010; Cook et al., 2015; Tejedor et al., 2016; Casas-Gómez et al., 2020), speleothem (Vansteenberge et al., 2016), ice cores, sediments (Luoto and Nevalainen, 2017) and documentary and instrumental evidence (Pfister et al., 1999; Brázdil and Dobrovolný, 2009; Dobrovolný et al., 2010; Wetter et al., 2011)."

43-44: To be reformulated, please!

**Author's Response:** "As an example of Hansson et al. (2011), which introduced a runoff series for the Baltic Sea only, between 1550 and 1995 using temperature and atmospheric circulation indices." has been changed to
**L39-41** ➜ "As an example, Hansson et al. (2011) introduced a runoff series for the Baltic Sea for the period 1550–1995 using temperature and atmospheric circulation indices."

47-48: To be reformulated, please!

**Author's Response:** "This can be achieved through a process-based model of varying complexity, with the advantage of following general physical laws – e.g., preserving mass balance, etc. Physical based models:"

**L47-52 ➜** "The available reconstructed precipitation and temperature series (or fields) can be used to reconstruct runoff with hydrological (process-based) models (Tshimanga et al., 2011; Armstrong et al., 2020) respecting general physical laws, such as preserving mass balance (e.g. MIKE SHE, Im et al., 2009; or VELMA Laaha et al., 2017) or data-driven methods which are able to capture complex non-linear relationships (for instance support vector machines Zuo et al., 2020; Ji et al., 2021; artificial neural networks ANNs; Senthil Kumar et al., 2005; Hu et al., 2018; Kwak et al., 2020; random forests Ghiggi et al., 2019; Li et al., 2021; Contreras et al., 2021)."

51: Reference Breiman et al., 2001 do not refer to a ML application for runoff/streamflow forecasting. You can find better ones

**Author's Response: L52 ➜** Reference was added: Li, Y., Wei, J., Wang, D., Li, B., Huang, H., Xu, B., Xu, Y. (2021). A Medium and Long-Term Runoff Forecast Method Based on Massive Meteorological Data and Machine Learning Algorithms. Water, 13(9), 1308. doi:10.3390/w13091308.

51: Reference Thiesen et al., 2019 and "shannon entropy" are not used for runoff/streamflow forecasting

**Author's Response: L52 ➜** Reference has been removed

53: Contrasting (changing) ➜ Changing

**Author's Response: L53 ➜** Text was updated

53: Suggestion: limit their application outside boundary conditions observed during model training.

**Author's Response:** Text has been modified as, **L52-54 ➜** "While, the lack of physical constraints in the data-driven models limits their application under changing boundary conditions (in comparison with those of the model training period), their advantage is that they can often directly use biased reconstructed data as an input series."

55: long-term ⟶ multi-century annual runoff reconstruction for 14 European catchments

**Author's Response: L55 ➜** Text has been updated
"The objective of the present study is to provide a multi-century annual runoff reconstruction for 14 European catchments,"

57: Remove: proxy data (if you follow Important Remark 3)

**Author's Response: L57 ➜** Text has been updated

57: "other long-term historical data sources" ➜ GRDC and scPDSI

**Author's Response: L57 ➔** Following text has been added "Old World Drought Atlas Self-calibrated Palmer Drought Severity Index (scPDSI) reconstruction (Cook et al., 2015). "

57: we use a combination of ➔ we benchmarked the use of

**Author's Response: L68-70 ➔** Text has been updated as,
"For validation the reconstructed datasets, we considered the observational data records of precipitation and temperature (Menne et al., 2018), as well as runoff from the Global Runoff Data Center (GRDC; Fekete et al. 1999), which was also used for model calibration."

58: Conceptual HM ⟶ Conceptual lumped HM

**Author's Response: L58 ➔** Conceptual HM is revised as "Conceptual lumped hydrological"

59: annual evolution ➔ annual runoff

**Author's Response: L60 ➔** Text has been updated

59: "We pay particular attention to low flows during drought years." The models are not optimized to pay particular attention to negative annual runoff anomalies so I would avoid such sentence.

**Author's Response: L60 ➔** Suggested statement has been removed

60: "Using long-term data on climatic conditions and runoff may provide an efficient technique of visualizing droughts and low flow periods". Please reformulate or remove.

**Author's Response:L60 ➔** Suggested statement has been removed

63: Drought identification ⟶ Drought identification methodology

**Author's Response: L62-63 ➔** Text has been updated

63. To be reformulated. Suggestion: The accuracy of the employed precipitation and temperature reconstructions, as well as the derived runoff simulations, is evaluated in Section

**Author's Response: L63-64 ➔** "The accuracy of the employed precipitation and temperature reconstructions, as well as the derived runoff simulations are evaluated in Section 4."

69: data from ➔ scPDSI drought indicator data from

**Author's Response: L68 ➔** Suggested term has been defined

69: Remove: natural proxies (if you follow Important Remark 3)

**Author's Response: L68 ➔** Text has been removed

75: To be reformulated, please!

**Author's Response:** "To this end Pauling et al. (2006), reconstructed precipitation (P) was done..."
The above line has been changed to
**L74 ➔** "Reconstructed precipitation ($P$) was derived by Pauling et al. (2006) through principal component regression..."

76. What about subparagraph: 2.1.1 Precipitation, 2.1.2 Temperature?

**Author's Response: L72-73 ➔** Subsection Precipitation has been introduced and the text "temperature gridded data" is removed.
**L77 ➔** Subsection Temperature has been added.
**L79-80 ➔** "Reconstructed temperature data was available in the same spatial and temporal resolution as precipitation."
**L80 ➔** The phrase "both of these" has been added.

94-96: Consistency: Choose between dataset or data-set

**Author's Response: L86, L89... ➔** Suggested term (dataset) has been revised.

104: "The runoff series from the GRDC were selected based on the condition of data availability, at least 25 years prior to 1900." ➔ Only GRDC runoff time series with at least 25 years of data prior 1900 were selected

**Author's Response: L97-98 ➔** The sentence was reformulated, as suggested above, so now it should be more clear.

124: Remove "we" ➔ Section 3.4 ...

**Author's Response: L117** has added 'Section 3.4' in the manuscript.

125: Section 3.5 presents the methods to identify annual runoff droughts

**Author's Response: L118 ➔** Suggested statement has been updated

132: and the proxy data and

**Author's Response: L123** ➔ The text "and the proxy data and" has been removed.

133: validation of individual catchments (Fig.2) ⟶ Fig 2 refers to P evaluation

**Author's Response: L124** ➔ It appears that Latex had a typos problem. It is now Table 2 at L124

135: See Important remark 4

**Author's Response: L128-129** ➔ Text has been removed and added the following line.
"The catchment average precipitation, temperature and scPDSI were estimated from the corresponding (gridded) data sets by averaging the relevant grid cells over the catchments."

181: Provide the metrics in Capital Case format

**Author's Response: L178-181** have been modified with Capital Case format.
"We used a set of seven statistical metrics to assess the performance of simulated runoff, namely: Nash–Sutcliffe efficiency (NSE), Pearson Correlation (R), Standard Deviation Ratio (rSD), Kling-Gupta efficiency (KGE), Root Mean Square Error (RMSE), Mean Absolute Error (MAE), Bias (BIAS) and Relative Bias (relBIAS). The mathematical formulations of these metrics are provided in Appendix A1."

199-204: This maybe belong more to Section 3.2 and 3.3

**Author's Response: L197-201** ➔ We keep the original text since an introduction sentence is essential to describe related analysis.

212: "Some stations indicated a worse performance and could not adequately capture the observed temperature variability".
➔ Very likely, is not the station that has bad skill, but the reconstruction
➔ "Low skill observed at some GHCN stations can be explained by the unresolved variability of grid-cell average temperature, especially in regions with complex terrain."

**Author's Response: L212-213** We agreed, the formulation was unfortunate. The statement was modified as suggested.
"Low skill observed at some locations can be explained by the unresolved variability of grid-cell average temperature, especially in regions with complex terrain."

217: Consistency: GOF or gof

**Author's Response: L213** ➔ This whole paragraph has been deleted because of aggregated time series analysis was already suggested to remove. But, we kept consistent (GOF) throughout the manuscript.

236-239: Move to Section 3.1

**Author's Response: L129-131 ➔** These lines are moved to section 3.1 and was deleted from L224.

240-241: GR1A is not driven by gridded data, but the catchment average value . . . ! Maybe move to Section 3.2.

**Author's Response:** Section 3.2 has been explicitly stated, as suggested.
**L136-137 ➔** "where $Q$, $E$ and $P$ represent annual runoff, basin average potential evapotranspiration and basin average precipitation, respectively and $i$ denotes the year."

260: Please reformulate (or remove between brackets content)

**Author's Response:** We removed the brackets' contents and replaced them with the following:
**L244-246 ➔** "Across many study locations, the combination of reconstructed forcings and their 1-year lag performed the best in terms of rapid convergence (the number of iterations needed) and high accuracy from all input combinations for both data-driven models (BRNN, LSTM)."

277: I don't get how scPDSI provide better representation of the temporal dependency structure

**Author's Response: L260 ➔** This statement regarding scPDSI dependancy was deleted.

287-288: Please reformulate

**Author's Response:** Following lines has been reformulated Eventually, we decided to utilize that model since the metrics used (NSE, KGE, R, D, RMSE, MAE) to produce better results in one particular model.
**L267-269 ➔** "Secondly, we identified the candidate best models for each of the 14 selected catchment, considering the GOFs based on the validation NSE and R greater than 0.5 and 0.7, respectively. The best model for each catchment was finally subjectively selected from those models considering the remaining validation measures (BIAS, rSD, KGE, RMSE and MAE) as well."

294: Please reformulate

**Author's Response:** The whole statement was revised.
**L274-277 ➔** "The latter figure compares the cumulative distribution functions of annual runoff for the periods 1500–1800, 1800–1900 and 1900–2000, as simulated by the BRNN(P+T+Lag) and LSTM(P+T+PDSI) – the two best performing models – and the GR1A (the most deviating simulation from the best model) with the distribution of the observed annual runoff for the Basel-Rheinhalle Rhine catchment."

297: simulations ➔ cumulative distribution of simulated runoff value

**Author's Response:**
**L278-279 ➔** Text has been revised as "The cumulative distribution of BRNN and LSTM simulated runoff values."

319: match, less ➜ agreement, lower

**Author's Response: L303**, text has been updated as "The agreement between the simulated and observed runoff deficit is lower compared to the annual runoff time series."

354, 358, 362,374,376: runoff ➜ annual runoff

**Author's Response: L354, L356, L361, L373, L375** are changed as annual runoff

359: conceptual ➜ conceptual lumped

**Author's Response: L357**, text has been updated

371-373: Maybe remove?

**Author's Response:** The text have modified a bit to explain the fact.
**L370-372** ➜ "Moreover, proxy records that were used for the derivation of precipitation and temperature input fields are spatially heterogeneous with some regions being better represented than others. This inevitably leads to poor performance over the latter."

374: develop ➜ derive

**Author's Response: L373** ➜ Term has been updated

396. Specify g and o before starting describing the metrics.

**Author's Response:** We chose p as a better symbol for the predicted value instead of g as gridded.
**L393** ➜ The terms $p_i$ and $o_i$ refer to the predicted and observed time series at point $i$ respectively.

395: measurement ➜ metrics

**Author's Response: L394** ➜ Text has been updated

396: ratio of standard deviations

**Author's Response: L395** ➜ Term has been changed as "Standard Deviation Ratio".

435: Remove: and epochs

**Author's Response: L433 ➜** Text has been removed

435-437: Suggestion: The Huber Loss is employed to minimize the mean absolute error between observations and predictions. Model checkpointing is used to keep track of model weights evolutions during training and select the best model weights when the allocated max number of training iterations is reached.

**Author's Response:** This comment was not entirely clear to us. The loss function (MAE) was in our case minimized during model compilation. However, we improved the description of the model setup and training. See methods and Appendices A2 and A3.

**References**

[revised manuscript text omitted]

---

## Author Comment (AC2)

**Reply to Reviewer 2**

We greatly appreciate the Reviewer efforts to evaluate our work, and we thank the Reviewer for the remarks that have allowed us to improve further on the presentation and clarify various parts of the previous edition, as described below.

**Reviewer 2**

"A 500-year runoff reconstruction for European catchments" by Sadaf Nasreen et al.

The manuscript "A 500-year runoff reconstruction for European catchments" by Sadaf Nasreen et al. shows the work and effort that has been done to create a new dataset of long-term runoff reconstruction for various European catchments. While reconstructions of meteorological variables such as temperature and precipitation were already available, this study closes the gap by providing open source runoff reconstructions. This is valuable information as it can provide historical context for upcoming studies, which are interested in assessing present and future extremes such as droughts. To create the runoff reconstruction, a semi-empirical hydrological model (GR1A) as well as two data-driven model (LSTM and BRNN) were used and tested for their suitability. An extensive data collection including precipitation, temperature, drought indices, natural proxy data and runoff observations over the period 1500 to 2000 were used to calibrate and validate the models. The data-driven models showed the most promising results, being able to correct for biases in the input data compared to semi-hydrological model. Furthermore, the separate analysis focussing on droughts showed that the reconstructed timeseries of these models correlated well with the historical documented droughts. The paper was well written and included extensive information on the approach and validation of reconstructed runoff timeseries, especially regarding drought events. The main points for improvement are mainly focussing on additional clarifications regarding certain aspects of the methods. Therefore, I would like to recommend publication after minor revisions. The comments for improvement can be found below.

**Major Comments**

Section 3.3 Data-driven models: While there is an extensive general explanation on the LSTM model, the BRNN description falls short. More importantly, information necessary to be able to follow as a reader on how the data driven models were trained and the final model parameters are not given or not fully clear. Aspects on the training/testing data, like the type of splitting or how much was withheld for training/testing purposes are not clear. Furthermore, aspects on the input data (for example the gridded+proxy, gridded+PDSI, and gridded+lag) and its preparation (e.g. type of normalization, handling with outliers, etc) would be of interest as well. An additional table (could be in the Appendix) with the input parameters as they are used in the ML models would support the readers understanding. While the Appendix covers the LSTM model structure, similar information on the BRNN is missing. Additional suggestion is to add the final model parameters (e.g. amount of neurons) into the schematization (Fig A.1) as well.

Overall I think adding more specific information on the ML models will only improve the readers understanding and as the data-driven models show the most promising results highlighting and clarifying their use is important.

**Author's Response:** We have indeed realised that the readers could be confused from the considered models and their description, especially without adding the information about training setup and tuning parameters. As a result, we considerably revised the main text to include the procedures for testing, training, and validation at the beginning of the methods section and improved the text flow. A section on general description of BRNN was added to Appendix A3. The model structure and parameter definitions are also included in the Appendix Table A1. We hope that it is now clear in the revised text.

Section 3 Methods: A schematic overview of the data-preprocessing (3.1), the incorporation of all the different datatypes and sources in the different model types (3.2 and 3.3) as well as the postprocessing steps (3.4 and 3.5) would be a nice addition to this section to not only visualize the general approach of the study but also support the subchapters and the readers understanding.

**Author's Response:**
We appreciated your suggestion. We agreed on this point in order to provide a clear representation of the workflow. We included Figure 2 in the manuscript to illustrate the work flow for data preprocessing, model selection and training technique selection as well as visualization methods.

**Minor Comments**

line 43: As an example: Hansson et al. . . . remove "of"

**Author's Response: L40 ➜** The text was removed.

Fig 1 (and also Fig 5 and Fig 6): think about changing red or green to a different colour to ensure that colorblind people can follow your figures.

**Author's Response:** According to the Reviewer suggestion, the colors in Figures 6 and 7 (previously labeled as Fig 5 and Fig 6) were modified. Additionally, the colors in Figure 1 were changed.

Line 75: sentence not flowing, for example move comma in front of reference and remove 'was done'

**Author's Response** This sentence was properly rephrased.
**L74-76 ➜** "Reconstructed precipitation (P) was derived by Pauling et al. (2006) through principal component regression to documented evidence (i.e., memoirs, annals, newspapers), speleothem proxy records (Proctor et al., 2000) and tree-ring chronologies from the International Tree-Ring Data Bank (ITRDB)."

Fig 2 and Fig 3 same range and colorbar per evaluation metric (makes it easier to compare), list min and max values of scale bar for readability

**Author's Response:**
**Figures 3 and 4 (updated figure numbers) ➜** "We have listed the minimum and maximum values for both metrics figures. Some metrics have varying ranges in both figures (For example; relBIAS and BIAS, RMSE for P and T are different), hence, we keep them as an original scale. The color bar in both figure's were made identical but with different ranges."

Section 3.4: possibly some lines on the pros and cons of the GR1A model

**Author's Response:**
**L140-142 ➜** "Compared to other conceptual models from the GR family (GR4J, GR5J), GR1A is simple to use, and allows for analyzing many variants, particularly defining best antecedent rainfall and potentially useful to predict the likelihood of floods and droughts (Mouelhi et al., 2006)."

Section 4.1 (and throughout the rest of the manuscript): be consistent in addressing GOF (now a mix between GOF and gof)

**Author's Response:** The GOF symbol was made consistent across the manuscript in response to the Reviewer remark.

Section 4.2 the information on calibration and validation should be part of the Methods and the models. Furthermore, it would be nice to move a figure with the time series of one station as seen in Fig S1 from the supplementary to that paragraph

to highlight calibration and validation periods.

**Author's Response:** The method section has been updated to better explain the calibration and validation phases.
**L129-130 ➜** "Data were split into two parts: calibration (1900–2000) and validation (<=1900) to assess the model's accuracy and to select an appropriate model."
In addition, time series of two best runoff reconstruction (Fig. 8) were included in the manuscript.

Line 255: 'greatly increased the performance (NSE from 0.2 to 0.62).' Compared to the values mentioned in prior example of Basel Reinhalle, 0.62 is not listed Table 3 for BRNN(Gridded+PDSI) but 0.57

**Author's Response:** You are right, there was a mistake that was corrected in the revised manuscript.

Table 3: highlighting the different performances is a nice feature and helps spotting important trends, however the darkest colours make it hard to read the values (same for tables in supplementary). Maybe also add a note in the table description what the colour indication means.

**Author's Response:** The colormap of Table 3 was updated as similar to Figures 2 and 3. Also, Table 3 was altered to Figure 4 when the legend color bar was added. Likewise, the Supplementary Tables were changed.

Both stations at Basel show higher correlation scores for validation than calibration. Ideas why this is the case?

**Author's Response: Table 3 ➜** We do not have any convincing explanation since for the data-driven methods, the calibration exhibits a higher correlation than the validation as expected.

Figure 7: Whitespace around the figure seems to be cut too narrow as the max value for station BaselRheinhalle-Rhine is cut off (130 instead of 1300)

**Author's Response: Figure 9 ➜** The figure was corrected in accordance with the suggestion.

Line 347 and Table 5: listing of years does not include 1724, which is also indicated in bold in Table 5.

**Author's Response: L324 ➜** Thanks, the year 1724 was added

Section 6: 'using the data set below' move below

**Author's Response:** The text was corrected.

Appendix: add references to equations and in text (easier to follow in case chapter layout changes)

**Author's Response:** Thank you, the references were added in equations and text.

---

## Referee Report (RR1)

[referee-annotated manuscript omitted]

---

## Referee Report (RR2)

[referee-annotated manuscript omitted]

---

## Author Response (AR2)

**Department of Water Resources and Environmental Modelling**
Czech University of Life Sciences
Kamycka 129, 165 21  Prague 6, Czech Republic
Phone: +420 224 382 147, Fax +420 234 381 854
e-mail: michalkova@fzp.czu.cz, www.fzp.czu.cz

Dated: 4/15/2022

Christof Lorenz

Editor-In-Chief

Earth System Science Data

Karlsruhe, Germany

Dear Dr. Christof Lorenz,

We'd like to express our gratitude to you and the Reviewers for your thoughtful comments on our article, "A 500-year annual runoff reconstruction for 14 chosen European catchments. "

Those comments are all valuable and very helpful for revising and improving our paper, as well as the important guiding significance to our researches. We carefully reviewed the comments and made changes that we hope will be accepted.

We have carefully addressed Reviewer #1 comments, particularly those regarding the proposed acronyms, which have been revised throughout the manuscript. All heatmaps were newly generated, taking into account Reviewer #1 colour choice selection. Additionally, the text was revised and polished as it is required.

The very useful and remarkable comments from Reviwer #3 is also higly appreciated. We have added the trend anlysis in the discussion part of the manuscript. we analysed the trends in the decadal runoff anomalies calculated from the reconstruction over several time periods. The reconstructed annual runoff for 1500-2000 for each catchment was first aggregated to 10-year time scale and divided by mean annual runoff. Our finding shows that there is no systematic trend throughout the whole 1500-2000 period. For a number of catchments there is a clear period of sustained above (Orsova-Danube and Dresden-Elbe) or below (Blois and Montjean Loire) average runoff during ca 1600-1800, while for the rest the persistence is clearly weaker although the low runoff signal is still visible (BaselRheinhalle, Baselschifflaende and Koln Rhine). Additionally, the linear trends calculated over 1500-2000 period are significantly

negative for 13 (out of 14) catchments but there are also periods when most of the catchments show increasing significant trend, for instance the trend is significantly positive for 12 catchments for the 1800-2000 period. Looking at the most recent 1950-2000 period, the trend is negative for seven catchments. Please note, that since the most recent period is not included in the developed reconstruction, any possible climate change impacts are difficult to interpret. Another suggestion from Reviewer #3 to determine the similarities between the developed reconstruction and precipitation/temperature reconstruction at high elevations and domain boundaries.

The answer to such a comment is that the low skill for some catchments cannot be attributed solely to bias in reconstructed precipitation and temperature (as described in Sect. 4.1), but rather to low station coverage in some (especially northern) parts of Europe, resulting in biassed basin-average precipitation and temperature estimates.

The following are the main corrections in the paper and point by point responses to the reviewer comments:

Yours Sincerely,
On behalf of all coauthors,
Sadaf Nasreen

**Reply to Reviewers**

Dear Dr. Ghiggi, thank you very much for your constructive and valuable comments. We revised the manuscript carefully and hope that it now satisfies the proposed requirements for successful publication.

**Reviewer 1**

The manuscript by Nasreen et al., investigate the reconstruction of annual runoff timeseries for 14 European catchments over the period 1500-2000. In the first part, the authors evaluate the validity of an existing precipitation and temperature reconstruction dataset against GCHN stations. In a second step, they evaluate the use of 2 data-driven models and a lumped hydrological model to predict annual runoff. In a third section, they provide an overview on years with low annual runoff occurred in the selected 14 European catchments during the last 500 years.

**Minor Comments**

1. In Figure 1 ensure that the color of the runoff station markers is different from the land mask color. In the first submission, it was green and was displaying nicely. You might also slightly reduce the marker size.

   **Author's Response:** The figure was revised and the color of runoff variable was changed into green, and the marker size was reduced.

2. Section 3.1 "Data preprocessing" still require some rearrangement.

   **Author's Response:** Several changes were made to the 'Data preprocessing' section in order to improve readability and coherency. There are a few phrases that were omitted:
   ➔ "we created two datasets "
   ➔ "models (the GR1A hydrologic model, the BRNN and LSTM data-driven models) that were used for 1500–2000 runoff simulation."
   And datasets were replaced with databases (catchment and observational databases). Additionally, the specified model names (GR1A, LSTM and BRNN) were eliminated from the section.

3. The workflow schema of Figure 2 requires a bit of polishing. Please address the following point:
   - What the "Testing = 25%" of calibration data" refers to? It's never mentioned in the manuscript.
   - You specify "check quality" for "Catchment dataset" and "Forcing validation dataset" but you not describe it the procedure in the paper. Either describe or remove it. . .
   - You specify QQplot but they are not presented in the manuscript.
   - Please use relBIAS instead of PBIAS for consistency across the manuscript.
   - Start with a Capital letter each bullet

   **Author's Response:** The 'check quality' was eliminated, and testing 25% percent is described in 'Method' section. QQplot was replaced by a scatter plot. relBIAS was replaced by PBIAS. In addition, the first letter of each bullet was capitalized.
   Line 174-175 ➔ The testing set was for each learning exercise extracted from the calibration data (1900-2000) as a random fraction (25%).

4. Ensure acronymic consistency across the text, figure and tables. For example, use [P,T, Lag] instead of the also appearing "P+T+Lag", (P+T+Lag), (P, T, Lag) and (P, T, Lag P, Lag T) (in Figure 2).

**Author's Response:** Thank you for the remarkable suggestions. All figures and tables captions were changed to include the proposed acronyms.

5. In Section 3.5 you mention that "annual drought duration and severity were then calculated" but you never present the results. Consider removing the sentence or present such results. Also, please clarify in the section how you computed the annual runoff/streamflow deficit. Please clarify the threshold quantile you selected also in the caption of Table 4.

**Author's Response:** The sentence "annual drought duration and severity were then calculated" was deleted. The section was introduced with the definition of runoff/streamflow deficit which is as follows,
The cumulative difference between runoff and the threshold was determined for each identified drought year, called as runoff/streamflow deficit. The threshold of 5% was chosen to represent extreme events during the last 500 years, and it was revised in Table 4.

6. Figure 3 and 4 can be further improved
- It looks really strange to me to see the minimum and maximum values of the stations skill to appear at the extrema of the colorbar. This is not common practice!
- You might want to consider centering the colormap for relBIAS around 0.
- You should enlarge the extent/bounding box of each plot to not mask some stations (i.e. in Sicily).
- Instead of repeating "Temperature"/" Precipitation" above each colorbar, put it as a title over the figure

**Author's Response:** The colorbar extrema in Figures 3 and 4 have been corrected. The colorbars on the precipitation and temperature maps are evenly spaced. When dealing with the relBIAS comment, the centre is already zero, and the scale smaller than zero has a maximum number of divisions. As a result, we chose not to have the colorbar uniformly spaced, but rather to use prime number spacing (7,5,3,2,1). As a result, the range of relBIAS values is -1.6 to 0.2. Furthermore, each plot enclosing box has been expanded to include a station in Sicily. Furthermore, the temperature and precipitation text above each colorbar has been eliminated, and is now only mentioned in the caption text.

7. Figure 5 would be easier to interpret with a diverging colorbar centered at the (NSE?) value that you consider satisfactory (i.e. blue-yellow-red colormap, with colors tending to red when below the satisfactory threshold).

**Author's Response:** The figure was modified and referred to as Table 3. Following the Ghiggi suggestion, the colormap was changed accordingly.

8. In the legend of Figure 6, it appears the term "gridded" that should be replaced by P, T to avoid confusion related to the nature of the model inputs.

**Author's Response:** To maintain consistency across the paper, the legend for Figure 5 (formerly Figure 6) was substituted with P, T.

9. In Figure 7, you should consider reducing slightly the size of the point markers. Also consider changing the color of the GRDC observations (i.e. in black) to highlight that the CDF is available only in the 1900-2000 period for Basel Rheinhalle-Rhine catchment.

**Author's Response:** According to Ghiggi recommendation, Figure 7 was adjusted. The size of the point markers was decreased, and the colour of the GRDC observations was changed to black.

10. In the text you refer as Figure 9 providing QQ-plots, while it contains scatterplots of modelled vs observed runoff. Please correct it.

**Author's Response:** The caption text of Figure 8 (formerly referred to as Figure 9) was updated.

11. In Sect 4.4 you refer to Figure 9 instead of Figure 10

   **Author's Response:** The figure reference was corrected.

12. In both Figure 9 and 10, please set the aspect ratio of each plot to 1 and ensure to set the same axis limits for the x and y axis. It helps in see under/overestimation patterns in the scatterplots! Please also consider reducing the marker size of the plot for an improved visualization.

   **Author's Response:** Both figures were updated with same axis limits and reduced the marker size. Since changing the aspect ratio to one shrinks the figure dramatically, we've decided to leave it at its original size.

13. Instead of discussing about minimum/maximum or low/high runoff deficit values, please use the terms small(est)/ large(st) runoff deficit.

   **Author's Response:** Terms were modified according to the proposed suggestions.

14. The definition of KGE in Appendix 1 is wrong.
   - (rSD 2) should be replaced with (rSD -1)
   - (beta 2) should be replaced with (beta 1).
   - Also note that (beta - 1) correspond to relBIAS

   **Author's Response:** Thanks for pointing out the mistake, The definition of KGE was updated.

L56 ➔ precipitation (P), temperature (T)

**Author's Response: L56 ➔** The text was updated.

L61-62 ➔ The structure of the paper is as follows: the considered hydroclimatic reconstructions, drought indicator and observed data are introduced in Sect. 2.

**Author's Response: L60-61 ➔** Sentence was modified as:
"Section 2 introduces P and T hydroclimatic reconstructions, the scPDSI drought indicator as well as precipitation, temperature and runoff observations."

L62 ➔ 'hydrological and data-driven' maybe not worth specifying here

**Author's Response: L61 ➔** It was removed.

L63 ➔ "precipitation and temperature reconstructions"

**Author's Response: L62 ➔** The text was updated as: "P and T reconstructions."

L67 ➔ Herein, we used precipitation (Pauling et al., 2006) and temperature (Luterbacher et al., 2004) reconstructions

**Author's Response: L66-67 ➔** Sentence was modified as per suggestion:
"This section present the data used in this study. To force the models, we investigate the use of precipitation (Pauling et al., 2006) and temperature (Luterbacher et al., 2004)."

L68-70 ➔ "For validation the reconstructed datasets, we considered the observational data records of precipitation and temperature (Menne et al., 2018), as well as runoff from the Global Runoff Data Center (GRDC; Fekete et al., 1999), which was also used for model calibration." Note that Menne et al., 2012 refers to GHCN-D. You used only GHCN-M right? So Menne et al., 2018

**Author's Response: L68-70 ➔** The above sentence was modffied as:
"For validating the runoff reconstructions, we used runoff from GRDC Fekete et al. (1999). The accuracy of atmospheric forcing reconstruction used as model input was assessed using the observational data records of P and T from the Global Historical Climatology Network (GHCN;Menne et al., 2018)."

L88-89 ➔ Remove "(....was used to verify the accuracy of the precipitation and temperature reconstructions."

**Author's Response: L88 ➔** The text was deleted.

L91 ➔ Reformulate: V4 version were included in the preliminary analysis

**Author's Response: L91 ➔** The following text was added in the manuscript.
"(Menne et al., 2012) were used to assess the reconstruction accuracy of the P and T fields as an input into the considered models"

L91 (Menne et al., 2012).

**Author's Response: L92 ➔** The reference was deleted

L92 found ➔ selected

**Author's Response: L92 ➔** The text was updated.

L116 ➔ Used hydrologic (Sect. 3.2) and data-driven models (Sect. 3.3) for runoff simulation are introduced in the second part.

**Author's Response: L116-117 ➔** The text was modified as Ghiggi suggestion.
"The hydrologic and data-driven models used to generate the runoff reconstructions are presented in Sect. 3.2 and 3.3 respectively."

Figure 2 ➔ Calculate P,T catchment average
—Associate GRDC runoff data
—Quality checks? Not described in the paper ... describe or remove ...
—Same for Forcing validation dataset
—Dataset is more raw data
—Database is more something processed one —caption: A schematic overview of the study work-flow.

**Author's Response:** The 'Quality checks' was eliminated, and the text was modified according to the proposed suggestions. In addition, the figure 2 caption was updated.

L120 ➔ Not scientific writing. "We prepared two datasets".
Describe instead the datasets/database you generated
You could provide them some names to facilitate their call across the text (i.e. CatchDB, ObsDB)

**Author's Response:** We appreciate Ghiggi comment regarding introducing the call names.
As the datasets are only mentioned once in the 'Data pre-processing' section, we just changed the dataset into the database to

keep clarity and readability. As a result, the text was altered as follows:

L121 ➜ Two databases were considered to analyse and develop the annual runoff reconstruction.

L124 ➜ Second database was further divided into two parts....

L124-125 ➜ Remove "(The GR1A hydrologic model, the BRNN and LSTM data-driven models)"

**Author's Response: L125** The text was deleted.

L125 ➜ Scientific writing !

Several input variables were considered for inclusion in ...

You need to clean out a bit the sentences here ...

Lot of sentence are already introduced in the previous sections ...

**Author's Response: L123-125** The paragraph was modified as,

"The second database was created as the basis for runoff reconstruction containing the observed runoff data for 21 selected catchments (Table 2) and the corresponding input variables of the models used to generate the multi-century runoff reconstructions. Several input variables were considered for inclusion in models, such as reconstructed precipitation and temperature and Old World Drought Atlas scPDSI."

L144 ➜ Remove "Data-driven methods"

**Author's Response: L144** The text was deleted

L151-155 ➜ Reformulate: reconstructed precipitation and temperature fields is referred to as [P,T]

P,T, PDSI

P and T forcing

Ensure to always use such acronym across the manuscript !

Maybe for consistency with tables and figures you could use [P,T,Lag]

Do not repeat this. Say instead: and therefore were not included in presented analysis.

**Author's Response: L152-156** The suggested acronyms were used in the whole manuscript and the following text was modified as:

"Specifically, the network using only reconstructed precipitation and temperature fields is referred to as $[P, T]$, the network with reconstructed forcing and OWDA scPDSI is termed as $[P, T, PDSI]$; and finally the network which includes 1-year lagged P and T forcing in addition to actual P and T is referred to as $[P, T, Lag]$. We also considered and explored lag times longer than 1 year. However the correlation between precipitation and runoff drops significantly at lag times longer than 1 year, and therefore were not included in presented analysis."

L158-159 ➜ Reformulate: In this structure, LSTM allows to learn a long-term dataset and controls the overfitting problem (Chen et al., 2020).

**Author's Response: L159** The following text was updated in the manuscript. LSTM is known for efficient simulation of time series with long-term memory (Van Houdt et al., 2020).

L160 "a given" ➜ the

**Author's Response: L160** The text was updated.

L167-169 implement the initial values of the ➔ initialize using
Reformulate: Initial weights are set up based on a prior distribution function during model training. By applying Bayesian formulation, weight parameters keep updating prior probability distribution to the posterior probability distribution.

**Author's Response: L165-171** The text was modified to simplify the description as:
BRNNs are based on the recurrent neural networks, which are often used to model time-series data (Wang et al., 2007), and extend them with Bayesian regularization (Okut, 2016) to account for uncertainty related to network parameters and input data (Zhang et al., 2011).

L184 hydrological droughts ➔ annual hydrological droughts

**Author's Response: L186** The text was changed.

L188-189 ➔ Annual drought duration and severity (the cumulative difference of runoff and the threshold) were then calculated for each identified drought year. — This results are not presented right? This should be removed... or eventually added as perspective in the conclusion ... the presented dataset can be used to investigate ...

**Author's Response:** The text was modified as:
**L189-190** ➔ "After that, the difference between runoff and the threshold was determined for each identified drought year, called as runoff deficit."
Furthermore, we added the following sentence in the conclusion section.
**L377-378** ➔ "the presented dataset can be used to investigate (...)"

L214 goodness of fit (GOF) ➔ Introduce the acronym

**Author's Response: L215** The term "goodness of fit (GOF)" was defined.

L224 Reformulate: gridded reconstruction of P and T

**Author's Response: L225** The text was changed as:
"(...)was driven by catchment average P and T and calibrated using observed annual runoff."

L228 Reformulate: These (relatively poorer catchment skills in northern Europe)

**Author's Response: L229-234** The text was changed as:
The catchments with relatively poor skills are located in northern Europe, which is in line with the previous findings by Seiller et al. (2012), who noted that the lumped hydrological models often exhibit larger uncertainties and fail to capture the extreme catchment values (both high and low) in those regions. The low skill for some of the catchments cannot be easily attributed only to bias in reconstructed precipitation and temperature (described in Sect. 4.1) but rather to low station and proxy coverage at some (especially northern) parts of Europe, leading to biased basin-average precipitation and temperature estimate.

L239 0.57 and 0.59 for validation ➔ maybe just report validation to simplify the reading?

**Author's Response: L244** The text was modified as:
"(...)(NSE 0.76 for calibration and 0.57/0.59 for validation, for BRNN/LSTM respectively)"

Caption Figure 5 satisfactory validation ➔ Maybe directly say: with NSE < 0.5 over the validation period !

**Author's Response:** The caption of Table 3 (previously named as Figure 5) was changed as:
"with NSE > 0.5 over the validation period".

L263 validation NSE of 0.5

**Author's Response: L268-269** The text was modified as: As a threshold, we considered validation NSE greater than 0.5 for at least one model.

L270 combination of reconstructed forcing with lagged values results ➔ models employing also reconstructed forcing with 1-year time lags results (...)

**Author's Response: L278** The following text was modified:
"The combination of reconstructed forcing with 1-year time lag results."

Figure 7 ➔ Put in black ...
➔ There are not enough GRDC data in 1800-1900 to build a meaningful CDF right.
➔ Reduce the size of the markers to increase plot clarity

**Author's Response:** According to Dr. Ghiggi recommendation, the colour of the observed GRDC data in Figure 6 (formerly referred to as Figure 7) was changed to black, and the legend acronyms [P,T], [P,T, Lag], and [P,T, PDSI] were modified. The marker sizes were also lowered.

L283 the models ➔ the models in the period 1500-1800 appears to be 1669 while in the past century (1900-2000) to be 1921.

**Author's Response: L293-294** The following text was updated:
"The models in the period 1500-1800 appears to be 1669 while in the past century (1900-2000) to be 1921. "

L287 I think it would be useful to note out that in the period 1500-1800 the CDF has very much lower/higher runoff values for BRNN and GR1A. LSTM seems to extrapolate less...
A comment of forcing uncertainty might be worth ..
GR1A simulate a minima of 250 mm/year of Rhine in Basel, while the the observed minima in the past century is above 500 mm/year ,,,

**Author's Response: L289-291** The following text was modified as:
Our finding shows that GR1A simulates a Rhine minima of 279 mm/year in Basel, whereas the observed minima in the past century is greater than 532.6 mm/year, inferring that CDF has significantly lower/higher runoff values between 1500 and 1800 for BRNN and GR1A, whereas LSTM appears to extrapolate less.

L289-290 ➔ Fig. 9 does not show the QQplot But predicted vs observed runoff !! QQ plots have 0-1 (units on both axis)

**Author's Response: L294-295** Thank you for pointing out the error, the text 'scatter plot' was updated.

L295 ➔ for the comparability

**Author's Response: L299** The following text was updated:
"to enable comparison (Supplementary Figs. S4 and S5)."

L296-297 ➔ the correlation

**Author's Response: L2307** The text was updated as: the correlation (reproduction of interannual dynamics).

L302 I have difficulty in observe this. You should put the axis of the plots in log scale !
And have the same x and y limits !!!!

**Author's Response:** There's no need for a log scale because it will transfer low flow values. While, the limits of axes were made equal.

L308 In the next step ➔ Use another word ....

**Author's Response: L319** The following word was updated: "Furthermore"

L310 maximum/minimum deficit ➔ I am not sure I understand what you refer to.
The minimum runoff value and the q0.05 value?

**Author's Response: L321** The text was updated as:
"the large deficit values for catchments (below 5th percentile)"

L312 ...also more severe in terms of hydrologic shifting. Please reformulate?

**Author's Response: L323-324** The following text was updated
"more severe in terms of changing hydrologic conditions."

L320 "maximum" ➔ maybe better to use the word "largest" deficit

**Author's Response: L331** The text was changed.

L321 severe ➔ extreme

**Author's Response: L326** The text was changed.

L322 Alternatively ➔ In the Koln-Rhine catchment , 26 remarkable ...
Why specific to a catchment in the middle of a general analysis

**Author's Response: L333** There was discussion of two significant Rhine river catchments in the manuscript, Basel Rheinhalle (1669) and köln (1686), both of which had the highest runoff deficits in the 17th century (see Table 3). The text was updated according to the suggestion.

L323 "In addition" ➔ The 1616 is considered the ...

**Author's Response: L334** The text was changed.

Figure 10 ➔ Same axis limits / ratio !!!

**Author's Response:** The axis was modified in response to Dr. Ghiggi suggestion.

L334 high ➔ large

**Author's Response: L345** The text was updated.

L336 including London ➔ why specific London ? Is southern England no? Or is where the analysis of Cook et al., 2015 has focused on?

**Author's Response: L334** The text was modified and the citation (Bonacina, 1923) was added. The OWDA map demonstrates the 1921 severe drought that hits southern England and central Europe, also detailed in (Bonacina, 1923).

L337 I would rewrite as:
The low river level of the Rhine, Thames and Loire river has been also documented by .... respectively
photographs from the De Telegraf documented all these rivers?
van der Schrier documented the Loire only?

**Author's Response: L350-352** Also reported in newspapers, The Rhine River (Switzerland), Molesey Weir, on the Thames River (United Kingdom), and Loire River (France) all have low river flows in 1921 (van der Schrier et al., 2021).

L339 "The precipitation totals were recorded as the lowest since 1774, and the year was also ranked top (in terms of deficit rainfall) in the Great Alpine region (Haslinger and Blöschl, 2017), where the rainfall deficit began in winter 1920/21 and lasted until autumn 1921."
This should be moved above after Cook et al., 2015 above when discussing of rainfall ...

**Author's Response: L348-350** The text was shifted.

L341-342 ...with some of our study catchment ➔ in agreement with some of our catchment reconstructions signaling the 1976 as a yearly drought in the ....

**Author's Response: L353-354** The text was modified accordingly.

L346 or other references ➔ and previous works

**Author's Response: L357** The text was modified accordingly.

L348 This might be the case as ➔ This sentence should be removed in my opinion. It appears like a surprise / just luck that your reconstruction agree with previous work ... :)

**Author's Response: L358-359** The text was updated as, "Because the tree-ring proxies involved in the developed reconstruction were the same, which could reveal the true nature of hydroclimatic shifts."

**Author's Response: L359** The text was deleted.

L348 developed reconstruction ➜ were used to derive the P+T reconstruction used to force our models ?

**Author's Response: L358** We used P, T, PDSI, and Lag-forced data to develop a runoff reconstruction

L349 Still ➜ Remove
It's important to note that the presented runoff reconstructions might have missed notably documented dry events.
Maybe I would add a word to the fact that analyzing year values, extreme dry summers could be compensated by very wet spring/autumn periods ... which mask the sub-yearly dry period. It's a problem of resolution, not methodology ...

**Author's Response: L362-366** The text was deleted and replaced with the following lines:
"It is important to note that the presented runoff reconstructions might have missed notably documented dry events." And the proposed reason was included in the concluding section at Lines 395-397. "Finally, since the runoff reconstruction is annual, the dry summers can be compensated by wet winters masking the sub-annual dry periods. However, this should be regarded as a resolution not methodology related problem."

L354 Following ➜ After careful validation

**Author's Response: L376** The text was replaced as: "After comprehensive validation of the simulated series, "

L354 we provided ➜ this work provides

**Author's Response: L377** The text was replaced.

L364 correlated ➜ correlates

**Author's Response: L387** The text was replaced.

Table 4 Minimum ➜ Largest

**Author's Response: Table 5** We decided to remove the "Minimum deficit (year)" column as it provided no meaningful information to the analysis. And, the text of table was changed and improved.

L366 series ➜ runoff

**Author's Response: L389** The text was replaced.

L374 are ➜ were

**Author's Response: L399** The text was replaced.

L374-375 (as was proven in validation).➜ Remove this is not necessary in the conclusion

**Author's Response: L399** The text was removed.

L394 deviation ➜ Deviation (SD)

**Author's Response: L419** The text was removed.

L396 with SD the standard deviation ➜ maybe not necessary ...

**Author's Response: L421** The text was removed.

L401 n ➜ I would put n out of the sum maybe

**Author's Response: L425** The term was modified.

L418 KGE 1? Why 2? Note that with your current formula (beta-1) ... is equivalent to relBIAS !
KGE is isually defined as mean/sd of o/p or p/o as in your case... ????

**Author's Response: L442** The term was modified. relBIAS took the place of $\beta - 1$, and 1 replaced by 2.

L424 Tensorflow capital case. If you want to keep Tensorflow ... add the reference Abadi et al. 2016

**Author's Response: L439** The following reference was added as: "Tensor flow (Abadi et al., 2016)."

L433-434 The checkpoint algorithm is also applied to test the model's accuracy level. Finally, the best output of the model is saved.

**Author's Response: L456-457** The text was modified as : "Model checkpoints is used to save the model having minimum loss during the training ... "

L444-445 After getting the optimal model, the data is further evaluated the performance on testing data and predicted runoff values for the previous 500 years. ➜ This does not belong to the appendix

**Author's Response: L467** The text was removed.

Table A1 Runoff ➜ runoff

**Author's Response: Table A1** The text was updated.

**Reply to Reviewer 2**

We greatly appreciate the Reviewer efforts to evaluate our work, and we thank the Reviewer for the remarks that have allowed us to improve further on the presentation and clarify various parts of the previous version. In the following, we provide detailed replies to all comments and discuss changes to the main manuscript. We hope that we have properly addressed all the comments and suggestions.

**Reviewer 2**

This paper reconstructed annual runoff for 14 European basins for the period 1500 to 2000. Pre-existing temperature and precipitation reconstructions were first assessed against observations. Two data-driven models and a lumped hydrological model were then used to predict annual runoff.

**Major comments**

In the Introduction, the authors mention that there has been a decease in annual river flow at some river gauges in Europe over the last decade by up to 22%. This is said within the context of climate change. The authors now have time series of annual runoff for several basins in Europe stretching from 1500 to 2000. It would be interesting to see if there are any trends in annual runoff over that period, and if the effects of climate change are evident in the latter part of the timeseries.

   **Author's Response:** Thank you for the remarkable comment. An additional trend analysis was performed and following text was added in the discussion part of the manuscript.
"Finally, we analysed the trends in the decadal runoff anomalies calculated from the reconstruction over several time periods. The reconstructed annual runoff for 1500–2000 for each catchment was first aggregated to 10-year time scale and divided by mean annual runoff. The resulting series are shown in Appendix Figure 1. It is clear that there is no systematic trend throughout the whole 1500–2000 period. For a number of catchments there is a clear period of sustained above (Orsova-Danube and Dresden-Elbe) or below (Blois and Montjean Loire) average runoff during ca 1600–1800, while for the rest the persistence is clearly weaker although the low runoff signal is still visible (BaselRheinhalle, Baselschifflaende and Koň Rhine). The linear trends calculated over 1500–2000 period (Table A2) are significant negative for 13 (out of 14) catchments but there are also periods when most of the catchments show increasing significant trend, for instance the trend is significant positive for 12 catchments for the 1800–2000 period. Looking at the most recent 1950–2000 period, the trend is negative for seven catchments. Please note, that since the most recent period is not included in the reconstruction any possible climate change impacts are difficult to detect."

[Figure]

**Figure 1.** Decadal fluctuation of runoff anomalies in selected catchments over the past 500 years.

**Table 1.** The average, minimum and maximum slope of the runoff anomalies over different periods.

| Time period | Number of catchments with (+/-) trends | Sign of trend | Average (min, max) slope |
|---|---|---|---|
| 1950-2000 | 7 | - | -2.573 (-4.374, -0.912) |
| 1950-2000 | 2 | + | 2.031 (1.447, 2.615) |
| 1900-2000 | 4 | - | -0.875 (-1.711, -0.480) |
| 1900-2000 | 4 | + | 0.664 (0.443, 0.852) |
| 1800-2000 | 2 | - | -0.419 (-0.540, -0.300) |
| 1800-2000 | 12 | + | 0.450 (0.066, 1.307) |
| 1700-2000 | 3 | - | -0.425 (-0.813, -0.045) |
| 1700-2000 | 10 | + | 0.388 (0.152, 1.288) |
| 1600-2000 | 6 | - | -0.317 (-0.859, -0.096) |
| 1600-2000 | 5 | + | 0.305 (0.045, 0.821) |
| 1500-2000 | 13 | - | -0.136 (-0.347, -0.046) |
| 1500-2000 | 0 | + | |

For the GR1A model, can you describe how X was optimized? Was the optimization manual or through some sort of algorithm?

**Author's Response:** For optimization of GR1A X parameter we used a gradient based minimization over a predefined interval as available in the AirGR package. This is now mentioned in the text.

Line 253: 'In the validation period, the differences between the models are more visible,'. Please discuss the possible reasons.

**Author's Response:** For the calibration period the models are optimized to match the observed runoff as much as possible. Therefore, provided the models are flexible enough and the data appropriate the simulated runoff should not differ dramatically between used models. In the validation period, given different input data the models are demonstrating their generalization skill which naturally differs from model to model. We added a remark on this.

In section 4.1 you mention that the reconstructions of temperature and precipitation differ from observations under certain circumstances: boundary of domain and high elevations. In section 4.3, you exclude the runoff simulations of seven basins due to the relatively poor performance of the models. I wonder if you can analyze if there are any similarities between the seven excluded basins. Since many basins have headwaters in mountainous regions, and since the reconstructions of rainfall and precipitation are relatively worse at high elevations, what does this mean for reconstructing annual runoff based on reconstructions of temperature and precipitation?

**Author's Response:** We added into Sect. 4.2, following explanation: The low skill for some of the catchments cannot be easily attributed only to bias in reconstructed precipitation and temperature (described in Sect. 4.1) but rather to low station coverage at some (especially northern) parts of Europe, leading to biased basin-average precipitation and temperature estimate.

**Minor comments**

Line 23: Water use by the power sector in the UK has dropped off considerably over the last 10 years as coal-powered plants are discontinued. I think maybe just revisit this statement for relevance over the last decade.

**Author's Response:** Thank you for this comment. Since the reference was not appropriate anyway, we dropped the sentence.

Figure 1: The color of the GRDC runoff gauges is too similar to the background map so they are hard to see. Throughout the paper, use a thousands separator and use the minus symbol rather than the hyphen when listing negative numbers. For example in Table 2.

**Author's Response:** The color of GRDC runoff gauges was changed to green and Table 2 symbols were updated.

Can Figures 3 and 4 be combined by using different shapes for the precipitation and rainfall symbols?

**Author's Response:** In the earlier stages of the manuscript preparation we have been considering this. However, the presented information was not clear enough (the symbols cannot be too small since their fill color gives the information on the error but then it leads to significant overplotting). Therefore we leave the presentation in two figures.

Line 242: ', the data-driven ? exhibited'

**Author's Response:** Text was updated as "data driven methods"

Figure 5 is actually a table. I would suggest formatting it as such. Also, for the 14 selected catchments with good GOFs in this table, can you group them together rather than using a bold outline.

**Author's Response:** In response to a reviewer suggestion, Figure 5 has been renamed Table, and the best ones have been grouped together in a rectangular block.

You mention a 'subjective' process in line 273. Can you somehow try and formalize the process of selecting the best model for each catchment?

**Author's Response:** We agree that more objective selection procedure would be beneficial. On the other hand, the performance of the best models was comparable and somewhat subjective choices (e.g. the considered metric, their number etc.) are unavoidable. Therefore we left this topic for future applications. We added a note in the manuscript: The best model for each catchment was finally selected from those models considering the remaining validation measures (relBIAS, rSD, KGE, RMSE and MAE) as well. Specifically, we picked the models with consistent good validation measures. This choice is partly subjective and more formal selection should be explored further. On the other hand, the candidate models were all performing comparably in most cases.

Figure 9 and 10: can you add the gradient of the line in each subplot as an indication of model bias?
**Author's Response:** The figures 8 and 9 were updated to include gradient lines.

**References**

Abadi, M., Barham, P., Chen, J., Chen, Z., Davis, A., Dean, J., Devin, M., Ghemawat, S., Irving, G., Isard, M., et al.: Tensorflow: A system for large-scale machine learning, in: 12th {USENIX} symposium on operating systems design and implementation ({OSDI} 16), pp. 265–283, 2016.

Bonacina, L.: The European drought of 1921, Nature, 112, 488–489, 1923.

Fekete, B. M., Vörösmarty, C. J., and Grabs, W.: Global, composite runoff fields based on observed river discharge and simulated water balances, 1999.

Luterbacher, J., Dietrich, D., Xoplaki, E., Grosjean, M., and Wanner, H.: European seasonal and annual temperature variability, trends, and extremes since 1500, Science, 303, 1499–1503, 2004.

Menne, M. J., Durre, I., Vose, R. S., Gleason, B. E., and Houston, T. G.: An overview of the global historical climatology network-daily database, Journal of Atmospheric and Oceanic Technology, 29, 897–910, 2012.

Menne, M. J., Williams, C. N., Gleason, B. E., Rennie, J. J., and Lawrimore, J. H.: The global historical climatology network monthly temperature dataset, version 4, Journal of Climate, 31, 9835–9854, 2018.

Okut, H.: Bayesian regularized neural networks for small n big p data, Artificial neural networks-models and applications, pp. 21–23, 2016.

Pauling, A., Luterbacher, J., Casty, C., and Wanner, H.: Five hundred years of gridded high-resolution precipitation reconstructions over Europe and the connection to large-scale circulation, Climate dynamics, 26, 387–405, 2006.

Seiller, G., Anctil, F., and Perrin, C.: Multimodel evaluation of twenty lumped hydrological models under contrasted climate conditions, Hydrology and Earth System Sciences, 16, 1171–1189, 2012.

van der Schrier, G., Allan, R. P., Ossó, A., Sousa, P. M., Van de Vyver, H., Van Schaeybroeck, B., Coscarelli, R., Pasqua, A. A., Petrucci, O., Curley, M., et al.: The 1921 European drought: Impacts, reconstruction and drivers, Climate of the Past Discussions, pp. 1–33, 2021.

Van Houdt, G., Mosquera, C., and Nápoles, G.: A review on the long short-term memory model, Artificial Intelligence Review, 53, 5929–5955, 2020.

Wang, W., Gelder, P. H. V., and Vrijling, J.: Comparing Bayesian regularization and cross-validated early-stopping for streamflow forecasting with ANN models, IAHS Publications-Series of Proceedings and Reports, 311, 216–221, 2007.

Zhang, X., Liang, F., Yu, B., and Zong, Z.: Explicitly integrating parameter, input, and structure uncertainties into Bayesian Neural Networks for probabilistic hydrologic forecasting, Journal of Hydrology, 409, 696–709, 2011.

---

## Author Response (AR3)

**Reply to Reviewers and Editor**

We sincerely thank the Editor and the Reviewers for providing very insightful comments. We have now addressed all the comments raised by the Reviewers and accordingly modified our manuscript. With this revision, we believe our manuscript is more focused and clear, and we are hopeful that it now satisfies the proposed requirements for successful publication. Additional minor modifications of typographical, and tense nature are made to the manuscript in order to improve its legibility. We would like to mention that after implementing all these changes in the revised manuscript and supporting information, the final findings and conclusions remain unchanged.

**Reviewer 1**

**Minor Comments**

L56 'the old'

**Author's Response:** The text was updated.

Table 1: Missing - at the end of the sentence

**Author's Response:** The caption text was updated.

Table 1: (CE) — relevant?

**Author's Response:** Yes, it is necessary to show common era time period .

L117 section ➜ Sect

**Author's Response:** The text was updated.

L118 section ➜ Sect

**Author's Response:** The text was updated.

L121 'forcing validation'

**Author's Response:** The text was modified as,
"—evaluating the accuracy of meteorological forcing reconstructions used for hydrological simulations"

L127 Second ➜ This

**Author's Response:** L128 ➜ The text was updated.

L155 presented ➜ in the presented

**Author's Response:** L156 ➜ The text was modified accordingly.

Figure 3 and 4 : Remove ➜ The left- and right- most figures represent the approximate minimum and maximum for the corresponding indicator.

**Author's Response:** In both figures, the following text in the caption was removed.

L210 Bias ➜ all uppercase

**Author's Response:** L211 ➜ The text was corrected.

L236 double space here?

**Author's Response:** The text was corrected.

L241 NSE 0.76 ➜ here missing the / other value

**Author's Response:** We just consider 0.76 because the results for LSTM and BRNN are identical for the case of $[P, T, PDSI]$.

L246 in ➜ in the

**Author's Response:** L247 ➜ The text was updated.

L276 are driving ➜ employed

**Author's Response:** L277 ➜ The text was updated.

L276 LSTM ➜ LSTM model

**Author's Response:** L277 ➜ The text was updated.

L277 was best in one case ➜ resulted the best in just one case

**Author's Response:** L278 ➜ The text was modified according to the Reviewer's suggestion.

L277 four ➜ timeseries reconstructions

**Author's Response:** L278-L279 ➜ The text was updated.

L291 Reformulate ➜ while in the past century (1900-2000) to be 1921

**Author's Response:** The text was modified as,
L292-293 ➜ "..., the year 1921 in the past century (1900-2000)"

L292-293 I would remove this. Or "worse" year is not appropriate term ...)

**Author's Response:** L294 ➜ The text was removed.

L302-305 ➜ I just noticed in Fig S5 an apparent bias of GRUN. I wonder if the difference is also due to the catchment polygon used to aggregate the GRUN runoff ... How did you delineate and validated the catchment out of curiosity?

**Author's Response:** During this process, each grid cell value was subjected to a time-series conversion of GRUN monthly runoff data into yearly scale. This annual runoff data has been spatially aggregated to include all grid cell values within each catchment polygon (shapefile). I agree with Ghiggi's observation that the resolution of GRUN data is 0.5 degree, which could result in a spatial mismatch across the catchment region, resulting in a slight discrepancy.

L355 Remove ➜ Because the tree-ring proxies involved in the developed reconstruction were the same, which could reveal the true nature of hydroclimatic shifts.

**Author's Response:** L356 ➜ The text was removed.

L360 analysed

**Author's Response:** The text was modified as,
"performed an exploratory analysis of decadal runoff ... "

L363 throughout ➜ in annual runoff !!!

**Author's Response:** The text was added.

L364 runoff ➜ annual runoff

**Author's Response:** L365 ➜ The text was added.

L376-377 Reformulate ➜ deficiencies in the driving input fields

**Author's Response:** The following text was added.
L372-373 ➜ "—there is high uncertainty in the forcing meteorological data. "

L378 the ➜ hydrological

**Author's Response:** L374 ➜ The text was added.

L386 runoff ➜ annual runoff

**Author's Response:** L382 ➜ The text was added.

L388 (GHCN) ➜ provided by GHCN station,

**Author's Response:** L384 ➜ The text was added.

L389 ed ➜ ion of the

**Author's Response:** L385 ➜ The text was replaced.

L389-390 I would remove this. You cannot assert inconsistencies in OBSERVED runoff based on the ESTIMATED RECONSTRUCTED FORCING

**Author's Response:** The following text was removed.
L386 ➜ "—leading to inconsistencies in observed runoff (e.g., demonstrated by the poor results of GR1A for some catchments)."

L393 remove "the"

**Author's Response:** L392 ➜ The text was removed.

L393 the ➜ years with
**Author's Response:** The text was changed.

L395-399 I would move this before discussing about the issue of the resolution of the reconstruction. Here you are still talking about uncertainties related to input data ....

**Author's Response:** The following text was moved.
L388-392 ➔ "The skill of precipitation and temperature reconstructions across the selected catchments to derive annual runoff is still fairly good. In addition, the data-driven methods that were used in the paper were capable of removing systematic bias. We cannot be sure, though, that the link between reconstructed forcing and annual runoff is stationary when going back in time. Moreover, when the number of natural proxies included in the derivation of the forcing dataset decreases, the uncertainty increases. The reconstructed data should, therefore, always be considered with caution."

L406 at free ➔ on the public

**Author's Response:** L401 ➔ The text was added.

L408 at website via link ➔ at

**Author's Response:** L403 ➔ The text was modified.

L409 ➔ I would remove all this. (GHCN) data can be accessed at ....

**Author's Response:** The following text was removed.
L404 ➔ "— provides revision and updated version (V4) for temperature and (V2) precipitation which"

L437 Replace 'predictions'

**Author's Response:** L431 ➔ The text was updated.

L453 Remove ➔ a function

**Author's Response:** L447 ➔ The text was updated.

Figure A2: Maybe could be nice to color bar above 0 in blue, and bar below 0 in red ;)

**Author's Response:** Thank you for your suggestion, the color bar was modified accordingly.

Table A2: Why there are always info for double periods? What they refer to? I don't understand the table Make no sense to me to report the slope statistics I would not add this to the manuscript. Out of scope.

**Author's Response:** The twofold period displaying both a negative and a positive trend for particular catchments. According to the Dr.Ghiggi recommendation, we have also realized that the table lacks relevancy; therefore, the table and its associated text have been deleted from the discussion section.

**Reviewer 3**

The only minor issue I could pick up is a slight inconsistency in tense in a few parts such as in Section 2 where there is a mix of past and present tense.

**Author's Response:** According to the Reviewer's recommendation. We carefully revised Section 2 content and and edited to maintain consistent grammatical tense.

---

## Author Response (AR4)

**Department of Water Resources and Environmental Modelling**
Czech University of Life Sciences
Kamycka 129, 165 21  Prague 6, Czech Republic
Phone: +420 224 382 147, Fax +420 234 381 854
e-mail: michalkova@fzp.czu.cz, www.fzp.czu.cz

Dated: 7/04/2022

Christof Lorenz
Editor-In-Chief
Earth System Science Data
Karlsruhe, Germany

Dear Dr. Christof Lorenz,

We'd like to express our gratitude to you for your thoughtful comments on our article, "A 500-year annual runoff reconstruction for 14 chosen European catchments.

In the trend analysis (Lines 360), all  unnecessary information was deleted,  and  the  text  was revised. Please note that in addition to the material presented in the paper we also used linear regression to assess the significance of the trends in annual runoff for several periods. However, those results were deleted in previous review round. We mention this in the text in the current version.

We believe that our manuscript is more focused and clearer now, and that it now satisfies the requirements for publication.

Yours Sincerely,
On behalf of all coauthors,
Sadaf Nasreen

---

## Author Response (AR5)

**Reply to Editor**

Dear Dr Christof, thank you very much for your constructive and valuable comments. We revised the manuscript carefully and it is worth mentioning here that Table 3 has been renamed to Figure 5 because all co-authors have agreed that heatmaps should be written in Figures as well as Supplementary Tables. We further hope that it now satisfies the proposed requirements for successful publication.

**Minor Comments**

L129 "<=" –> before
**Author's Response:** The text was updated.

L157 –> Either use Fig. or Figure throughout the text.
**Author's Response:** Thank you for your suggestion. Fig. was updated in the whole manuscript.

L159 LSTM –> It
**Author's Response:** The text was updated.

L164 by means of –> using/applying the
**Author's Response:** The text was updated.

L164 by means of –> using/applying the
**Author's Response:** The text was updated.

L361 time scale –> averages
**Author's Response:** The text was updated.

L362 Remove "Appendix"
**Author's Response:** The following text was removed.

L363 Rather –> Instead
**Author's Response:** The following text was replaced.

L365 ca –> approx.
**Author's Response:** The following text was replaced.

L366 Either Köln or Collogne
**Author's Response:** The following text was updated throughout the manuscript.

L366 significant –> significantly
**Author's Response:** The following text was replaced.